# Partial inhibition of mitochondrial complex I ameliorates Alzheimer's disease pathology and cognition in APP/PS1 female mice

Andrea Stojakovic[1,15], Sergey Trushin[1,15], Anthony Sheu[2,15], Layla Khalili[1], Su-Youne Chang[3,4], Xing Li[5], Trace Christensen[6], Jeffrey L. Salisbury[6,7], Rachel E. Geroux[1], Benjamin Gateno[1], Padraig J. Flannery[1], Mrunal Dehankar[5], Cory C. Funk[8], Jordan Wilkins[1], Anna Stepanova[9], Tara O'Hagan[9], Alexander Galkin[9], Jarred Nesbitt[1], Xiujuan Zhu[1], Utkarsh Tripathi[1], Slobodan Macura[7], Tamar Tchkonia[10], Tamar Pirtskhalava[10], James L. Kirkland[10], Rachel A. Kudgus[11], Renee A. Schoon[11], Joel M. Reid[11], Yu Yamazaki[12], Takahisa Kanekiyo[12], Song Zhang[13], Emirhan Nemutlu[14], Petras Dzeja[13], Adam Jaspersen[6], Ye In Christopher Kwon[2], Michael K. Lee[2,16] & Eugenia Trushina[1,11,16✉]

Alzheimer's Disease (AD) is a devastating neurodegenerative disorder without a cure. Here we show that mitochondrial respiratory chain complex I is an important small molecule druggable target in AD. Partial inhibition of complex I triggers the AMP-activated protein kinase-dependent signaling network leading to neuroprotection in symptomatic APP/PS1 female mice, a translational model of AD. Treatment of symptomatic APP/PS1 mice with complex I inhibitor improved energy homeostasis, synaptic activity, long-term potentiation, dendritic spine maturation, cognitive function and proteostasis, and reduced oxidative stress and inflammation in brain and periphery, ultimately blocking the ongoing neurodegeneration. Therapeutic efficacy in vivo was monitored using translational biomarkers FDG-PET, $^{31}$P NMR, and metabolomics. Cross-validation of the mouse and the human transcriptomic data from the NIH Accelerating Medicines Partnership–AD database demonstrated that pathways improved by the treatment in APP/PS1 mice, including the immune system response and neurotransmission, represent mechanisms essential for therapeutic efficacy in AD patients.

A list of author affiliations appears at the end of the paper.

Alzheimer's disease (AD) is a multifactorial disorder without a cure. It is characterized by progressive accumulation of aggregated amyloid β (Aβ) peptides and hyperphosphorylated Tau protein, memory decline, and neurodegeneration. The consistent failure of clinical trials focused on reducing Aβ levels and aggregation suggests that such therapies may not work in AD patients regardless of disease stage, underscoring the need to discover novel targets and therapies for AD[1–3]. Recent studies demonstrated that altered energy homeostasis associated with reduced cerebral glucose uptake and utilization, altered mitochondrial function and microglia and astrocyte activation might underlie neuronal dysfunction in AD[4–8]. Intriguingly, accumulating evidence suggests that non-pharmacological approaches, such as diet and exercise, reduce major AD hallmarks by engaging an adaptive stress response that leads to improved metabolic state, reduced oxidative stress and inflammation, and improved proteostasis[9]. While mechanisms of the stress response are complex, AMP-activated protein kinase (AMPK)-mediated signaling has been directly linked to the regulation of cell metabolism, mitochondrial dynamics and function, inflammation, oxidative stress, protein turnover, Tau phosphorylation, and amyloidogenesis[10]. Combined analysis performed using multiple types of genome-wide data identified a predominant role for metabolism-associated biological processes in the course of AD, including autophagy and insulin and fatty acid metabolism, with a focus on AMPK as a key modulator and therapeutic target[11]. However, the development of direct pharmacological AMPK activators to elicit beneficial effects has presented multiple challenges[12]. We recently demonstrated that mild inhibition of mitochondrial complex I (MCI) with the small molecule tricyclic pyrone compound CP2 blocked cognitive decline in transgenic mouse models of AD when treatment was started in utero through life or at a pre-symptomatic stage of the disease[13,14]. Moreover, in neurons, CP2 restored mitochondrial dynamics and function and cellular energetics. However, it was unclear whether MCI inhibition would elicit similar benefits if administered at the advanced stage of the disease, after the development of prominent Aβ accumulation, brain hypometabolism, cognitive dysfunction, and progressive neurodegeneration. As a proof of concept, we demonstrate that partial inhibition of MCI triggers stress-induced AMPK-dependent signaling cascade leading to neuroprotection and a reversal of behavior changes in symptomatic APP/PS1 female mice, a translational model of AD. Beneficial effect of treatment could be monitored using translational biomarkers currently utilized in clinical trials.

## Results

### CP2 activates AMPK-dependent neuroprotective pathways and restores cognitive and motor function in symptomatic APP/PS1 mice

The tricyclic pyrone, CP2, specifically inhibits the activity of MCI in human and mouse brain mitochondria[13] (Fig. 1a–c). CP2 penetrates the blood-brain barrier (BBB) and accumulates in mitochondria, mildly decreasing MCI activity, which leads to an increase in AMP/ATP ratio and AMPK activation[13]. CP2 was effective in blocking cognitive dysfunction when treatment was administered to pre-symptomatic mice carrying familial mutations in the APP(K670N/M671L) and PS1 (M146L) genes (referred to as APP/PS1 mice)[13,14]. To determine whether CP2 could engage AMPK-dependent neuroprotective mechanisms (Fig. 1d) in symptomatic mice, we administered a single oral dose to 9-10-month-old APP/PS1 mice and examined the expression of key proteins in each pathway after 4, 24, 48, and 72 h (Fig. 1e–l, Supplementary Figs. 1a, 2). An independent cohort of CP2-treated APP/PS1 mice was assayed using in vivo

18F-fluorodeoxyglucose positron emission tomography (FDG-PET) and compared to non-transgenic (NTG) untreated littermates (Fig. 1h, Supplementary Fig. 1b). Consistent with previous observations, CP2 robustly activated AMPK after 24 h, when increased phosphorylation of acetyl-CoA carboxylase 1 (ACC1), a biomarker of AMPK target engagement associated with increased fatty acid oxidation[15], was evident 4 h after CP2 administration (Fig. 1f, Supplementary Figs. 1a, 2). Remarkably, a substantial increase in the glucose transporters, Glut 3 and 4, was observed as early as 4 h after CP2 treatment that persisted for 24 and 48 h, consistent with the established role of AMPK in the maintenance of glucose uptake in the brain[16–18] (Fig. 1g, Supplementary Figs. 1a, 2). Improved glucose uptake in APP/PS1 mice was independently established using in vivo FDG-PET (Fig. 1h, Supplementary Fig. 1b). A decreased ratio of phosphorylated vs. total pyruvate dehydrogenase (PDH) at 24 and 48 h confirmed that augmented glucose uptake was associated with improved glucose utilization (Fig. 1g, Supplementary Figs. 1a, 2), since increased PDH activity leads to acetyl-CoA production from pyruvate in the glucose catabolism pathway, promoting energy production in mitochondria[19]. Furthermore, increased expression of the transcriptional coactivator, peroxisome proliferator-activated receptor-γ coactivator-1α (PGC-1α), mitochondrial transcription factor A (TFAM), and mitochondria-specific neuroprotective Sirtuin 3 (Sirt3), supports the view that there was improved mitochondrial biogenesis and function (Fig. 1i, Supplementary Figs. 1a, 2)[20,21]. We further confirmed that CP2 induces an anti-inflammatory mechanisms evident by increased levels of IκBα and decreased phosphorylation of p65, which together block NF-κB transcription factor activation (Fig. 1j, Supplementary Figs. 1a, 2)[22]. In addition, CP2 activates nuclear factor E2-related factor 2 (Nrf 2)-dependent expression of antioxidants including heme oxygenase (HO1), superoxide dismutase (SOD1), and catalases (Fig. 1k, Supplementary Figs. 1a, 2)[23]. Finally, CP2 treatment increased autophagy based on changes in the autophagy activating kinase ULK1, Beclin1 and LC3B proteins essential for autophagosome formation (Fig. 1l, Supplementary Fig. 1a, 2)[24]. These data demonstrate that oral CP2 administration activates the neuroprotective AMPK signaling network in symptomatic APP/PS1 mice at therapeutic doses.

To determine the therapeutic efficacy of chronic administration, we treated symptomatic APP/PS1 female mice with CP2 (25 mg/kg/day in drinking water ad lib) from 9 until 23 months of age (Fig. 2a). NTG female age-matched littermates were controls. All treated mice tolerated CP2 well; they did not manifest side effects and gained weight throughout the duration of the study with an increase in lean and fat mass (Fig. 2b, Supplementary Fig. 3a–l). Data generated using indirect calorimetry (CLAMS) demonstrated that CP2-treated and vehicle-treated NTG mice had similar food intake (Supplementary Fig. 3b). Metabolic rate at rest was decreased by CP2 treatment only in NTG mice (Supplementary Fig. 3c). Since general daily activity of vehicle-treated and CP2-treated NTG mice was not different (Fig. 2f), the significant weight gain observed in CP2-treated NTG mice could be associated with reduced metabolic rate and lower energy expenditure. In comparison to NTG mice, vehicle-treated APP/PS1 mice displayed lower body weight, higher food intake, increased metabolic rate and motor activity (Supplementary Fig. 3a–c, Fig. 2f). This is consistent with previous findings showing that AD mice have elevated caloric intake, hypermetabolism and lower body weight[25]. The weight gain in CP2-treated APP/PS1 could be explained by increased food intake and attenuated hyperactivity, while metabolic rate remained similar between CP2-treated and vehicle-treated groups (Fig. 2b,f, Supplementary Fig. 3a–c). Since CP2 treatment did not increase

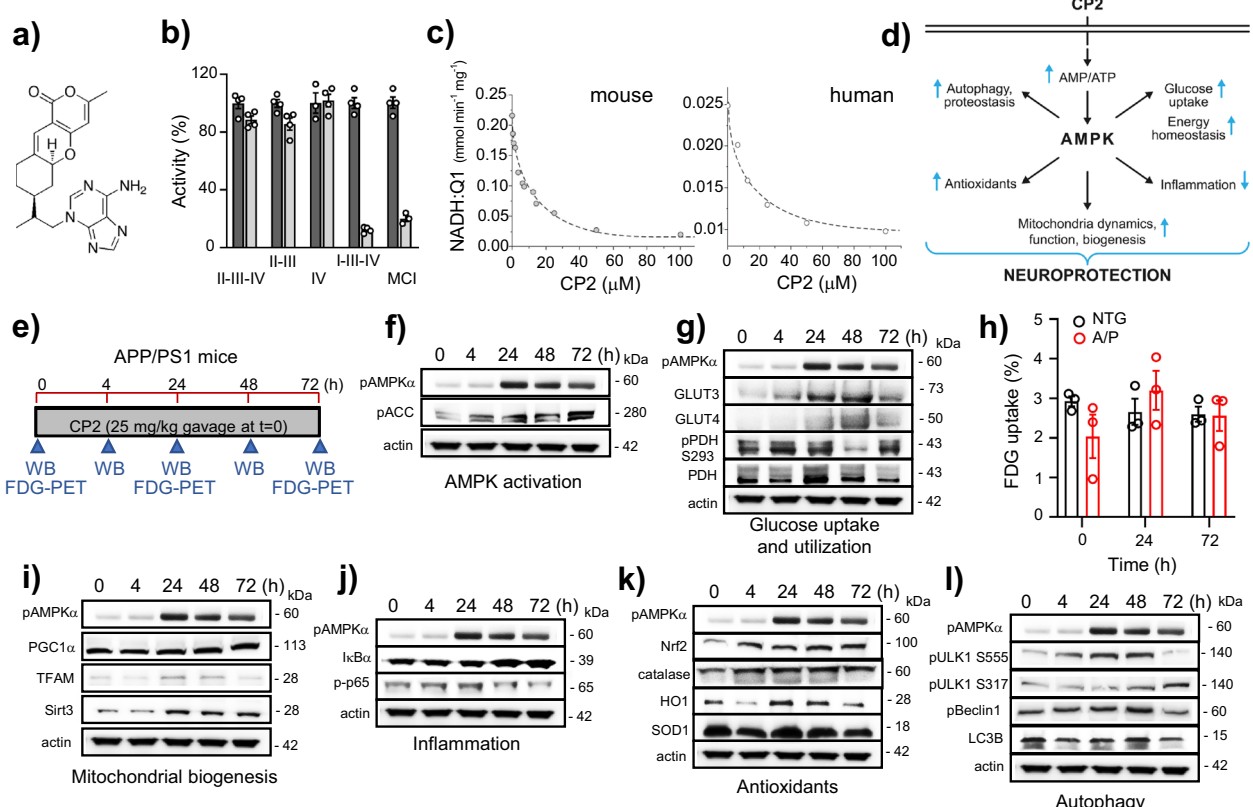

**Fig. 1 Acute CP2 treatment activates multiple AMPK-dependent neuroprotective mechanisms in symptomatic APP/PS1 mice improving energy homeostasis in the brain. a** CP2 structure. **b** CP2 (light gray bars, 50 μM) does not affect the activity of succinate oxidase (complexes II-III-IV), succinate: cytochrome *c* reductase (complexes II-III), and ferrocytochrome *c* oxidase (complex IV only), but significantly inhibits MCI affecting NADH oxidase (complexes I-III-IV) and NADH:ubiquinone (complex I only) in mouse brain mitochondria. Vehicle-dark gray bars. **c** CP2 inhibits MCI in mitochondria isolated from the mouse and postmortem human cortical tissue. **d** Neuroprotective pathways activated by CP2 in the brain converge on AMPK activation. **e** Timeline of acute CP2 administration via gavage (25 mg/kg) to APP/PS1 mice 9–10-month-old. Brain tissues from 1 mouse per each time point were examined using western blot analysis. Independent cohort of mice was subjected to the in vivo FDG-PET (*n* = 3 mice per group). **f–l** Western blot analysis of brain tissue from acutely gavaged APP/PS1 mice from **e** confirms that CP2 activates multiple AMPK-dependent neuroprotective mechanisms in symptomatic APP/PS1 mice. **h** Untreated symptomatic APP/PS1 mice 9–10-month-old have reduced glucose uptake in the brain determined using in vivo FDG-PET compared to NTG littermates (time 0 h). Acute CP2 oral administration improves glucose uptake brining glucose levels in APP/PS1 mice to the same level as in untreated age-matched and sex-matched NTG mice (time 24 and 72 h); *n* = 3 mice per group.

food intake in NTG mice, it is feasible that increased food intake in CP2-treated APP/PS1 mice could be related to more frequent feeding due to ameliorated hyperactivity.

Results of behavioral tests confirmed that CP2-treated APP/PS1 mice had improved spatial memory and learning (Morris Water Maze, Fig. 2c, d), restoration of attention and non-spatial declarative memory (Novel Object Recognition, Fig. 2e), reduced hyperactivity in the open field test (Fig. 2f), and increased strength and motor coordination (rotarod, hanging bar, Fig. 2g, h). CP2 penetrates the BBB[13,14] and has oral bioavailability of 65% (Supplementary Fig. 4). CP2 concentrations measured in the brain at the end of the study averaged at ~62 nM (Supplementary Table 1), consistent with earlier bioavailability studies[13,14]. To establish CP2 selectivity and specificity, we conducted in vitro pharmacological profiling against 44 human targets, including G-protein-coupled receptors, ion channels, enzymes, neurotransmitter transporters, and 250 kinases (Supplementary Tables 2–3, Supplementary Data 1). At 1 and 10 μM, concentrations higher than those found in the brain tissue of chronically treated mice (Supplementary Table 1), CP2 had minimal off-target activities demonstrating its selectivity at therapeutic doses. These observations reveal that chronic treatment with MCI inhibitor CP2 improves cognitive and motor function in APP/PS1 mice to that of NTG mice after the onset of significant accumulation of Aβ

plaques[26], and development of behavioral[27] and mitochondrial dysfunction[28].

**CP2 treatment improves glucose uptake and utilization and metabolic flexibility in symptomatic APP/PS1 mice.** We next determined the effect of chronic CP2 treatment on brain energy homeostasis. Similarly to AD patients, glucose utilization in the brain of APP/PS1 mice measured using FDG-PET imaging was significantly reduced (Figs. 1h, 3a, b). Consistent with results of the acute administration (Fig. 1h), chronic CP2 treatment for 1 - 2.6 months alleviated pronounced brain glucose hypometabolism in APP/PS1 mice (Fig. 3a, b). Data generated using CLAMS provided further evidence that, compared to untreated counterparts, CP2 treatment in APP/PS1 mice increased carbohydrate oxidation and metabolic flexibility, an essential ability to switch between lipid and carbohydrate oxidation that is affected in metabolic diseases and aging (Fig. 3c–e). Consistent with the improved regulation of glucose metabolism, CP2-treated APP/PS1 and NTG mice had decreased fasting plasma insulin levels and better insulin sensitivity and glucose tolerance (Fig. 3f–h). Western blot analysis in brain tissue revealed increased expression of Glut 3 and 4 and a decreased ratio of pPDH/PDH indicative of improved glucose uptake and utilization together with

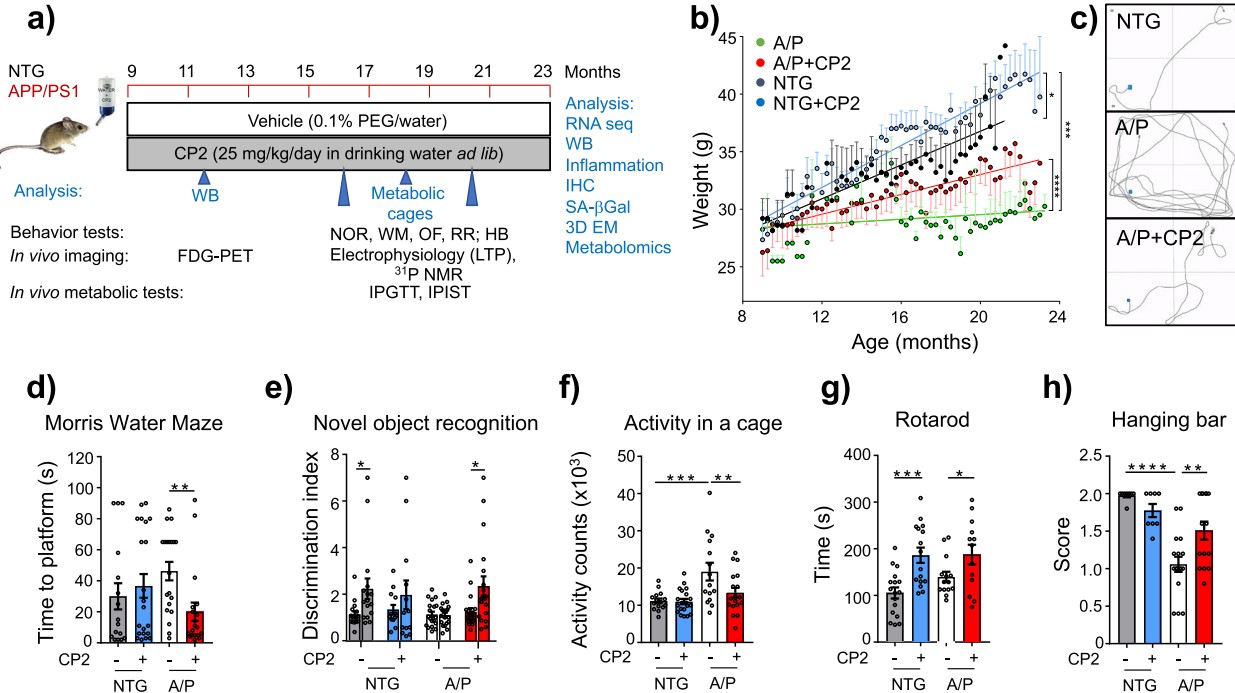

**Fig. 2 CP2 treatment restores cognitive function in symptomatic APP/PS1 mice. a** Timeline of chronic CP2 treatment. **b** Weight of NTG and APP/PS1 mice treated with vehicle or CP2 through the duration of the study. **c–h** CP2 treatment improves a performance in the Morris Water Maze (**c**, **d**) and in the Novel Object Recognition test (**e**), reduces hyperactivity of APP/PS1 mice in the open field (**f**), and increases motor strength and coordination on the rotating rod (**g**) and hanging bar (**h**). NTG, $n = 16$ mice per group; NTG + CP2, $n = 20$ mice per group; APP/PS1, $n = 21$ mice per group; APP/PS1 + CP2, $n = 19$ mice per group. Data are presented as mean ± S.E.M. Data were analyzed by two-way ANOVA with Fisher's LSD post-hoc test. A paired Student $t$-test was used for statistical analysis of NOR test. $*P < 0.05$, $**P < 0.01$, $***P < 0.001$, $****P < 0.0001$. Body weight data in **b** were analyzed by linear regression analysis: Linear regression indicates that slopes between NTG ($R$ squared $= 0.6821$) and NTG + CP2 ($R$ squared $= 0.9181$) are different ($P = 0.0234$); slopes between APP/PS1 ($R$ squared $= 0.04326$) and APP/PS1 + CP2 ($R$ squared $= 0.7668$) are different ($P < 0.0001$); slopes between NTG ($R$ squared $= 0.6821$) and APP/PS1 ($R$ squared $= 0.04326$) are different ($P = 0.0005$). In all graphs: APP/PS1, orange line; NTG, black line; NTG + CP2, blue line; APP/PS + CP2, red line.

enhanced signaling through the IGF pathway (Fig. 3i, Supplementary Figs. 5, 6).

Since CP2 inhibits MCI, we examined if chronic treatment affects ATP levels in the brain using $^{31}$P nuclear magnetic resonance ($^{31}$P NMR) spectroscopy (Fig. 3j, k). This method allows non-invasive measurement of in vivo energy metabolite concentrations including phosphocreatine (PCr), inorganic phosphates (Pi), and the α, β, and γ phosphate groups of ATP (Fig. 1j). Chronic CP2 treatment over 10 months did not decrease the PCr/ATP ratio in APP/PS1 or NTG mice (Fig. 3k) consistent with improved glucose uptake/utilization. As a second measure, we performed metabolomic profiling in brain of vehicle-treated and CP2-treated APP/PS1 and NTG mice treated with CP2 for 6 months (Supplementary Table 4). In agreement with target engagement, CP2 treatment increased levels of AMP in APP/PS1 mice but did not reduce brain levels of ATP. Treatment resulted in increased levels of citrate and N-acetyl aspartate (NAA), markers of improved mitochondrial and neuronal function. Importantly, levels of 2-hydroxyglutarate, a marker of detrimental mitochondrial stress[29], were not elevated, suggesting inhibition of MCI with CP2 does not induce adverse mitochondrial stress that could negatively regulate neuronal survival and function. Increased levels of gamma-aminobutyrate, a metabolite involved in the gamma-aminobutyric acid (GABA) neurotransmitter system, indicate an improvement in neurotransmission and potential reversal of AD-related neurodegeneration. Increased levels of ascorbic/dehydroascorbic acids imply an improvement in vitamin C status and redox balance in the brain, consistent with CP2-induced activation of neuroprotective mechanisms (Fig. 1d–l).

**CP2 treatment reduces Aβ-related pathology, inflammation, and oxidative stress and improves proteostasis in brain and periphery.** Histological examination of the hippocampus and cortex of CP2-treated APP/PS1 mice using 4G8 antibody revealed a significant reduction in Aβ plaques compared to untreated littermates (Fig. 4a, b). Biochemical analysis conducted using sequential extraction of brain tissue showed that CP2 treatment reduced total Aβ levels (Fig. 4c). While soluble Aβ was increased (Fig. 4d, e), levels of insoluble peptides were significantly decreased (Fig. 4f). Improved proteostasis could be promoted by autophagic degradation associated with AMPK activation and inhibition of the activity of glycogen synthase kinase-3 (GSK3β), whose hyperactivation in AD is directly linked to Aβ pathology[30,31]. Indeed, CP2-dependent AMPK activation increased inhibitory phosphorylation of GSK3β and promoted the expression of proteins associated with lysosomal biogenesis and autophagy, including the transcription factor EB (TFEB), lysosomal-associated membrane protein 1 (LAMP-1), and microtubule-associated protein light chain 3 (LC3B) (Fig. 4g, Supplementary Figs. 7, 8). These data further support the contention that CP2-induced autophagy is one of the neuroprotective pathways essential for Aβ clearance (Fig. 1d).

Since AMPK activation is known to reduce inflammation and promote anti-oxidant response[32], we examined levels of glial fibrillary acidic protein (GFAP) and the ionized calcium-binding adapter molecule 1 (Iba1), well-established markers of glial activation and inflammation, in the brain of CP2-treated and vehicle-treated APP/PS1 mice (Fig. 4h–k). Blood from the same

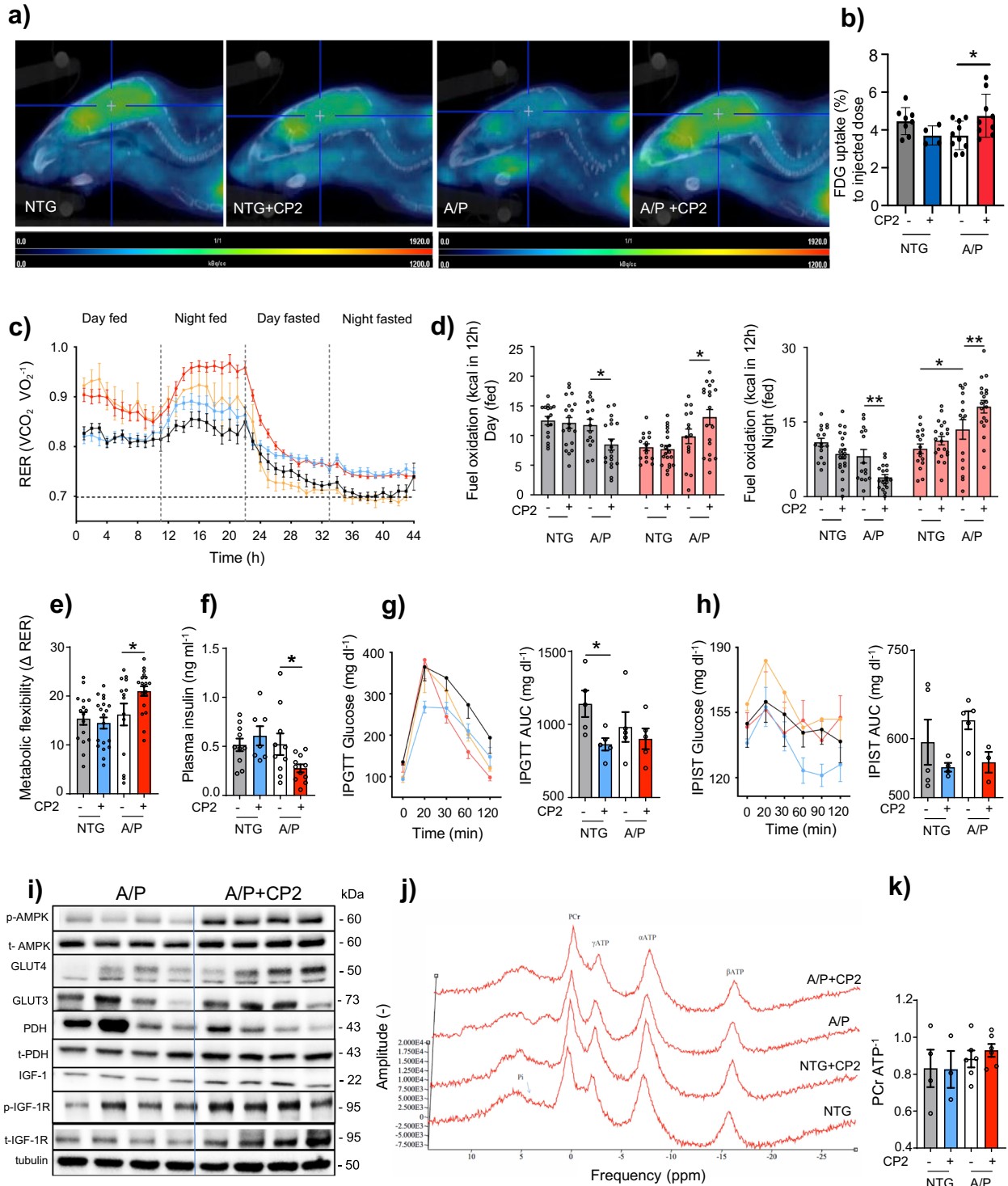

mice was profiled for cytokines and chemokines (Fig. 4l–o, Supplementary Fig. 9a). We found that treatment significantly reduced inflammation in the brain and periphery, decreasing glial activation (Fig. 4h–k) and pro-inflammatory markers (e.g., IL-12, TNFα, G-CSF, Fig. 4l–o). We next measured lipid peroxidation in the brain, a well-established marker of oxidative stress prominent in the hippocampus and cortex of AD patients, which correlates with the extent of neurodegeneration and Aβ deposition[33]. After 13 months of CP2 treatment, levels of malondialdehyde (MDA) were significantly reduced in both APP/PS1 and NTG mice (Fig. 4p).

Oxidative stress and inflammation induce cell proliferation arrest and the cell senescence phenotype, contributing to age-related diseases. Therefore, we examined levels of senescent cells in inguinal (ING) and periovarian (POV) adipose tissue from NTG and APP/PS1 mice using β-Galactosidase (β-Gal) staining[34] (Supplementary Fig. 9b–d). CP2 treatment significantly reduced abundance of senescent cells in both APP/PS1 and NTG mice. In AD patients, increased levels of ceramides, especially Cer16, Cer18, Cer20, and Cer24, were directly linked to oxidative stress and Aβ pathology[35]. Targeted metabolomic profiling conducted in plasma of CP2-treated mice revealed a significant decrease in

**Fig. 3 CP2 treatment increases glucose uptake and utilization in symptomatic APP/PS1 mice. a** Glucose uptake was increased in the brain of CP2-treated APP/PSI mice measured using FDG-PET after 1 - 2.6 months of treatment in two independent cohorts of mice. Representative images are from one of the cohorts. **b** Quantification of glucose uptake by FDG-PET imaging from **a**. NTG, $n = 7$ mice per group; NTG + CP2, $n = 4$ mice per group; APP/PS1, $n = 5$ mice per group; APP/PS1 + CP2, $n = 10$ mice per group. **c** Changes in respiratory exchange rate (RER) recorded in all treatment groups over 44 h during ad lib fed and fasting states in mice treated for 9 months. **d** Glucose oxidation was increased in CP2-treated APP/PS1 mice fed ad lib based on CLAMS data from **c**. Gray bars indicate fat consumption; orange bars indicate carbohydrate and protein oxidation. **e** Metabolic flexibility is increased in CP2-treated APP/PS1 mice based on their ability to switch from carbohydrates to fat between feeding and fasting states. **c–e** NTG, $n = 16$ mice per group; NTG + CP2, $n = 20$ mice per group; APP/PS1, $n = 15$ mice per group; APP/PS1 + CP2, $n = 19$ mice per group. **f–h** CP2 treatment reduces fasting insulin levels in plasma of APP/PS1 mice (**f**); increases glucose tolerance in NTG mice measured by intraperitoneal glucose tolerance test (IPGTT) (**g**); and displays tendency to improve intraperitoneal insulin sensitivity test (IPIST) in NTG and APP/PS1 mice (**h**) after 9–10 months of treatment. $n = 5$ mice per group. The outliers in (**h**) included 1 NTG+CP2 and 2 APP/PS1+CP2 mice that were excluded from the graph. **i** Western blot analysis conducted in the brain tissue of APP/PS1 mice treated with CP2 for 13 months indicates increased IGF-1signaling, expression of Glut 3 and 4 transporters and changes in pyruvate dehydrogenase (PDH) activation associated with glucose utilization in the TCA cycle. **j** Representative [31]P NMR spectra with peaks corresponding to energy metabolites, including inorganic phosphate (Pi), phosphocreatine (PCr), and three phosphate group peaks for ATP generated in living NTG and APP/PS1 mice after 9 months of vehicle or CP2 treatment. **k** Phosphocreatine/ATP ratio calculated based on the [31]P NMR in vivo spectra from **j**. NTG, $n = 4$ mice per group; NTG + CP2, $n = 3$ mice per group; APP/PS1, $n = 6$ mice per group; APP/PS1 + CP2, $n = 6$ mice per group. Data are presented as mean ± S.E.M. Data were analyzed by two-way ANOVA with Fisher's LSD post-hoc test. *$P < 0.05$, **$P < 0.01$. In all graphs: APP/PS1, orange line; NTG, black line; NTG + CP2, blue line; APP/PS1 + CP2, red line.

concentrations of Cer16, Cer18, and Cer24, specifically in APP/PS1 mice (Fig. 4q–t). These data suggest that CP2 treatment induces multiple protective mechanisms including autophagy, anti-inflammatory and anti-oxidant responses, which contribute to improved proteostasis, reducing Aβ pathology, which in turn could be monitored using a translational metabolomic approach.

**CP2 treatment improves synaptic function, long term potentiation (LTP), dendritic spine maturation, and mitochondrial dynamics.** Synaptic loss is the best correlate of cognitive dysfunction in AD[36]. To determine whether augmented cognitive performance after CP2 treatment was associated with improved synaptic function, we analyzed excitatory postsynaptic potential (fEPSP) in the CA1 region in acute hippocampal slices of APP/PS1 and NTG mice measuring local field potential (Fig. 5)[37]. We initially recorded basal synaptic transmission and strength of post-synaptic responses to electrical stimulation of Schaffer collaterals (Fig. 5a). Activation of Schaffer collaterals revealed reduction of fiber volley amplitudes in APP/PS1 mice compared to NTG mice (Fig. 5b, c), which was partially restored by CP2 treatment (Fig. 5c).

Since short-term plasticity plays a crucial role in neuronal information processing relevant to cognitive function, we next investigated the effect of CP2 on Schaffer collaterals-CA1 short-term plasticity utilizing a paired-pulse stimulation protocol. Paired-pulse facilitation (PPF) measures the ability of synapses to increase transmitter release upon the second of two closely spaced afferent stimuli, which depends on residual calcium levels in the presynaptic terminal[38]. If LTP is mediated presynaptically, an increase in transmitter release is accompanied by a change in short-term plasticity. We found that the PPF was not different between groups (Fig. 5d), suggesting that CP2-dependent improvement in LTP in APP/PS1 mice was not associated with the pre-synaptic release of neurotransmitters. We further applied tetanic stimulation to Schaffer collaterals-CA1 to induce and record LTP over 60 min to determine EPSP. Significant currents associated with strong LTP were recorded in NTG and CP2-treated NTG and APP/PS1 mice, while vehicle-treated APP/PS1 mice did not exhibit significant LTP formation (Fig. 5e, f). These data indicate that cognitive impairment in APP/PS1 mice could be associated with the inability to form and maintain LTP in the hippocampus, while CP2 treatment corrected this defect.

LTP critically depends on the morphology of dendritic spines, which determines synaptic strength and plasticity[39,40]. We examined dendritic spine morphology in the CA1 hippocampal region of vehicle-treated and CP2-treated APP/PS1 and NTG

mice using three-dimensional electron microscopy (3D EM) reconstruction (Fig. 6a, b)[41]. In NTG mice, the majority of spines were mature (thin, stubby, mushroom, and branched), while immature filopodia and long thin spines were prevalent in APP/PS1 mice (Fig. 6b, c). CP2 treatment promoted maturation of dendritic spines in APP/PS1 mice (Fig. 6b, c) and markedly improved spine geometry (the length and width of spine necks and heads, and compartmentalization factor) in NTG and APP/PS1 mice, indicating greater ability to maintain LTP[42] (Supplementary Fig. 10). In APP/PS1 mice, CP2 treatment restored the mushroom spine volume, length, and head width to the dimensions observed in NTG mice (Supplementary Fig. 10a–e). In NTG and APP/PS1 mice, CP2 significantly increased the compartmentalization factor (Supplementary Fig. 10f), a measure of the spine head depolarization during synaptic transmission, which is regulated by the length of the spine neck and indicates greater synaptic plasticity[43]. Increased spine maturation resulted in more active synapses in CP2-treated APP/PS1 mice, bringing synaptic activity to the level of NTG mice (Fig. 6d). Improved synaptic function in CP2-treated APP/PS1 mice was associated with increased levels of synaptophysin, the postsynaptic density protein PSD95, and BDNF (Fig. 6e–h, Supplementary Fig. 11). Thus, CP2-dependent cognitive protection is associated with improved morphology of dendritic spines, LTP, and synaptic transmission in the hippocampus.

Since synaptic activity requires energy, we examined mitochondrial integrity in the brain tissue of the same APP/PS1 and NTG mice utilized in the study of dendritic spines (Fig. 6k–n). Consistent with reports on mitochondrial fragmentation in AD, we observed increased number of round-shaped organelles in vehicle-treated APP/PS1 mice. CP2 treatment resulted in a higher number of elongated organelles and increased levels of neuroprotective Sirt3 in APP/PS1 mice (Fig. 6e, i, k–m), which could additionally contribute to reduced inflammation and neuroprotection[20,44,45]. Interestingly, the increased mitochondrial mass was observed only in CP2-treated NTG mice (Fig. 6n). Together with our previous report of CP2-dependent restoration of axonal trafficking and enhanced bioenergetics[13], these data demonstrate an improvement in mitochondrial dynamics and function in symptomatic APP/PS1 mice, which is essential for synaptic function and improved energy homeostasis.

**CP2 treatment attenuates the ongoing neurodegeneration.** Neurons of the locus coeruleus (LC) provide norepinephrine to the hippocampus, mediating memory and attention[46]. AD patients exhibit early neurodegeneration in the LC where

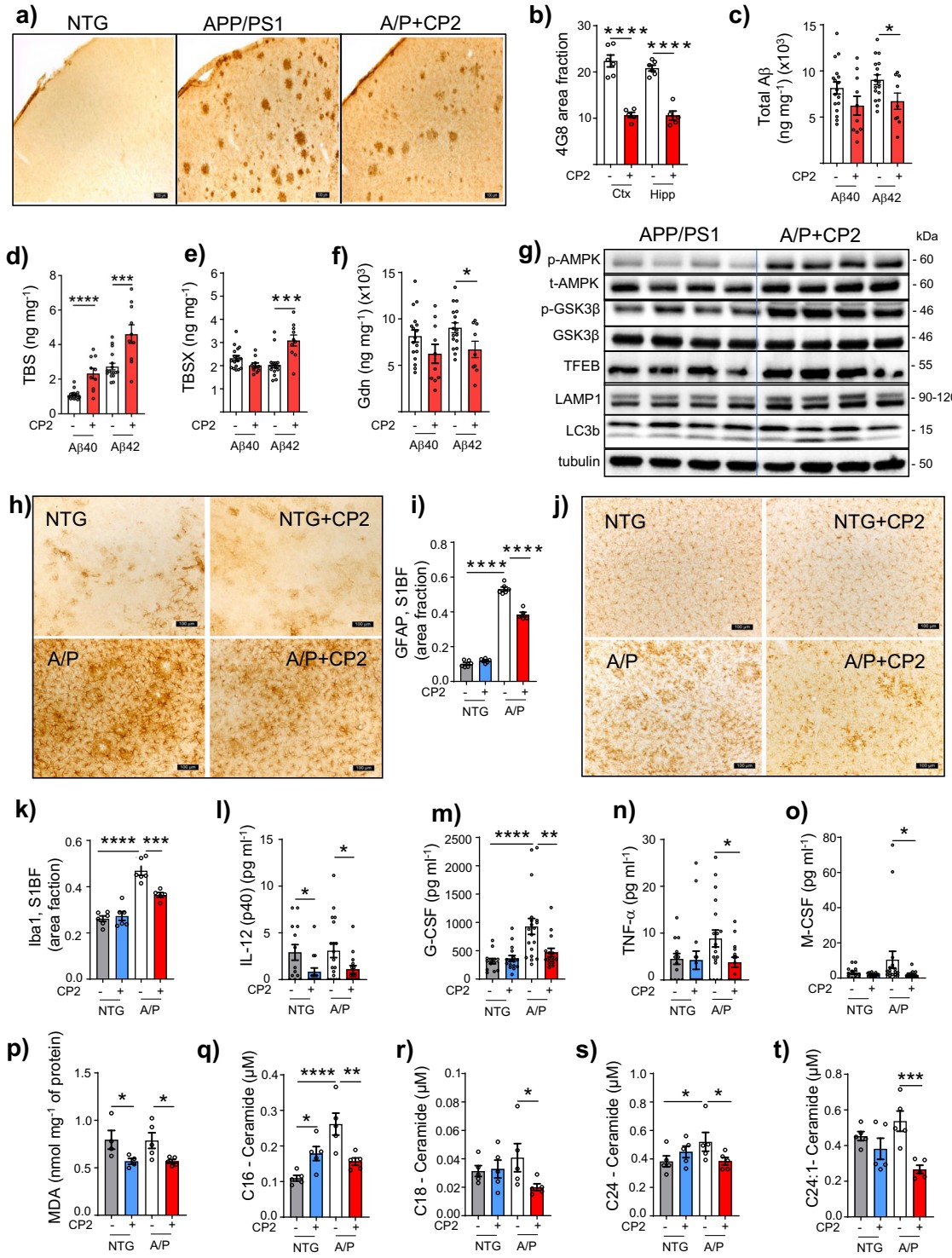

severity of neuronal loss correlates with the duration of illness[47]. Neurodegeneration in the LC has been shown to affect Aβ and Tau aggregation, inflammation, synaptic function, neuronal metabolism, and BBB permeability[48]. Previously, we demonstrated that the degeneration of LC neurons in human AD is recapitulated in mouse models of cerebral amyloid[49,50]. In the current APP/PS1 mice, progressive loss of TH+ cortical afferents starts at 6 months of age (Fig. 7a, b; Supplementary Fig. 12a, c), followed by the loss of noradrenergic (TH+) neurons in the LC (Fig. 7c, d, Supplementary Fig. 12b, d). Consistent with the progressive loss of afferents, there was a

reduction in the volume of TH+ neurons in APP/PS1 mice starting at 12 months of age (Fig. 7e, f). Analysis of 12 month old APP/PS1 mice that received CP2 treatment for 2 months (from 10 months of age) showed that CP2 did not impact neurodegeneration, since the cortical TH+ axon density, the number of TH+ neurons, and neuronal volumes were similar between vehicle-treated and CP2-treated subjects (Fig. 7b, d, f; 12-month-old group). In mice receiving CP2 for 10 months, further progression of neurodegeneration was completely halted by the CP2 treatment. Thus, cortical TH+ axon density, TH+ neuron number in LC, and TH+ neuronal volume in 20 month

**Fig. 4 CP2 treatment reduces levels of Aβ, inflammation and oxidative stress in symptomatic APP/PS1 mice. a** Representative images of Aβ plaques visualized using 4G8 antibody in *S1BF* cortex of NTG and APP/PS1 mice. Scale bar, 100 µm. **b** Levels of Aβ plaques are significantly reduced in *S1BF* and hippocampus in CP2-treated APP/PS1 mice estimated using 4G8 antibody shown in **a**. APP/PS1 (Ctx), $n = 6$ mice per group; APP/PS1 + CP2 (Ctx), $n = 5$ mice per group; APP/PS1 (Hipp), $n = 6$ mice per group; APP/PS1 + CP2 (Hipp), $n = 5$ mice per group. **c–f** Differential centrifugation and ELISA revealed decreased levels of total Aβ42 in brain homogenates from CP2-treated APP/PS1 mice (**c**). Levels of soluble Aβ40 and 42 obtained using TBS (**d**) and TBSX (**e**) fractions were increased, while concentrations of the least soluble Aβ40 and 42 were decreased in brain fractions obtained using Gdn (**f**). **c–f** APP/PS1, $n = 17$ mice per group; APP/PS1 + CP2, $n = 10$ mice per group. **g** CP2 treatment induces AMPK activation, reduces the activity of GSK3β, and activates autophagy in brain tissue of APP/PS1 mice. Each lane represents an individual mouse. **h, j** Representative images of GFAP + (**h**) and Iba1 + (**j**) staining in the *S1BF* in vehicle and CP2-treated NTG and APP/PS1 mice. Scale bar, 100 µm. **i, k** Quantification of GFAP (**i**) and Iba1 (**k**) staining from **h** and **j**, respectively. NTG, $n = 6$ mice per group; NTG + CP2, $n = 6$ mice per group; APP/PS1, $n = 6$ mice per group; APP/PS1 + CP2, $n = 4$ mice per group. **l–o** CP2 reduces pro-inflammatory markers in plasma of NTG and APP/PS1 mice. NTG, $n = 13$ mice per group; NTG + CP2, $n = 17$ mice per group; APP/PS1, $n = 20$ mice per group; APP/PS1 + CP2, $n = 16$ mice per group. **p** Levels of lipid peroxidation measured using MDA assay were significantly reduced in brain tissue of CP2-treated NTG and APP/PS1 mice. NTG, $n = 4$ mice per group; NTG + CP2, $n = 4$ mice per group; APP/PS1, $n = 5$ mice per group; APP/PS1 + CP2, $n = 5$ mice per group. **q–t** Concentrations of ceramides (C16, C18, C24, C24-1) were significantly reduced in blood collected from CP2-treated APP/PS1 mice and measured using targeted metabolomics. $n = 5$ mice per group. All mice were 23-month-old. Data are presented as mean ± S.E.M. A two-way ANOVA with Fisher's LSD post-hoc test was used for data analysis. For the comparison between vehicle and CP2-treated APP/PS1 groups (**b–f**), an unpaired Student *t*-test was used for statistical analysis. \*$P < 0.05$; \*\*$P < 0.01$; \*\*\*$P < 0.001$; \*\*\*\*$P < 0.0001$.

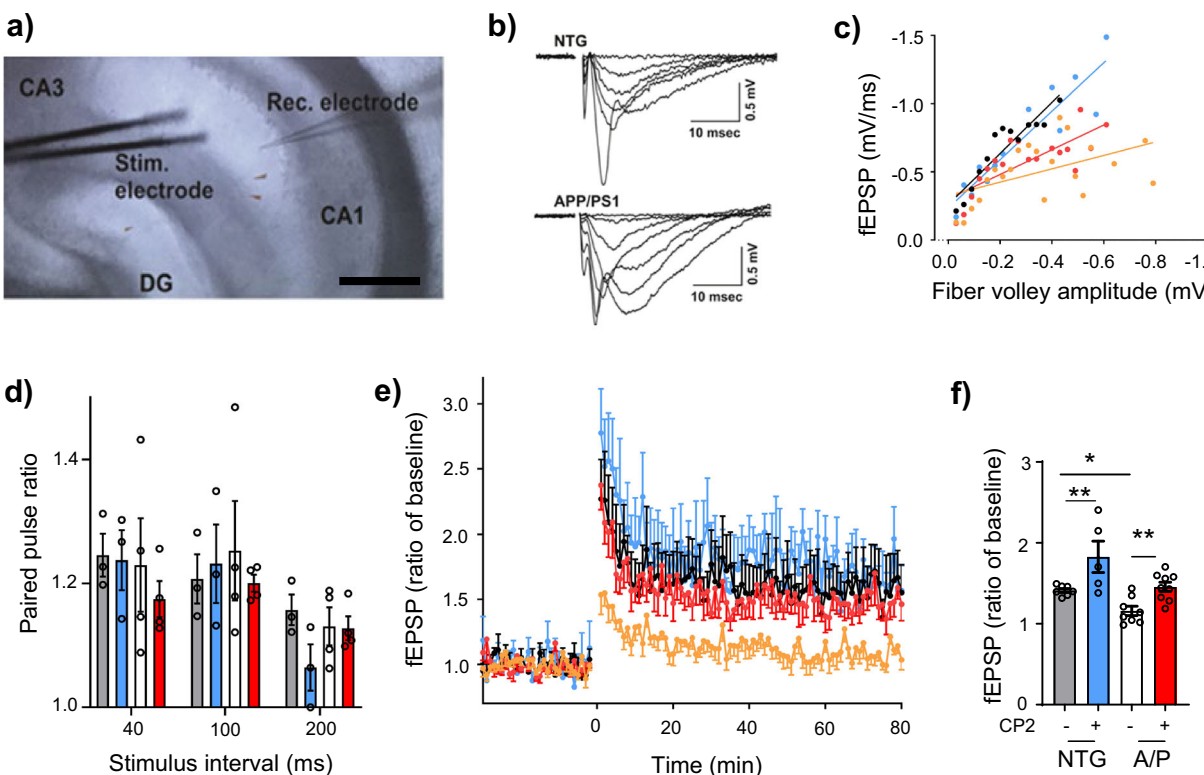

**Fig. 5 Chronic CP2 treatment improves synaptic activity and LTP in APP/PS1 mice. a** Experimental setting for stimulation-evoked local field potential (LFP) measurement in the hippocampal slice. The stimulation (Stim) electrode was placed at the Schaffer collaterals and the recording (Rec) electrode was placed at the stratum radiatum in the CA1. Scale bar, 200 µm. **b** Representative raw traces of fEPSP from NTG and APP/PS1 mice. Various stimulation intensities (10–300 µA) were applied to evoke fEPSP. The stimulation pulse width and intervals were fixed at 60 µs and 30 s, respectively. As the stimulation intensity was increased, the initial slope of fEPSP was increased. **c** CP2 effect on basal synaptic strength. To examine the pre-post synaptic relationships, initial slopes of fEPSP were plotted against amplitudes of presynaptic fiber volleys. Pre-post synaptic relationship in the CP2-treated APP/PS1 group was improved compared to the APP/PS1 group. **d** Paired-pulse facilitation did not differ between experimental groups. Two stimulations were applied with a short interval to determine presynaptic involvement in synaptic plasticity. **e** CP2 treatment improves LTP formation. Average traces for fEPSPs in hippocampal slices from each experimental group ($n = 2$–3 slices from 5 mice per group). Traces represent mean ± S.E.M. per time point. To induce LTP, three tetanic stimulations (100 Hz, 60 µs-pulse width for 1 s) were applied with 3-sec intervals. In APP/PS1 hippocampus, the tetanic stimulation induced early phase post-tetanic potentiation; however, long lasting potentiation was not observed. In the slices from APP/PS1 CP2-treated mice, LTP was induced and maintained over 60 min. NTG, black line; APP/PS1, orange line; NTG + CP2, blue line; APP/PS1 + CP2, red line. **f** LTP intensities among groups were compared at 60 min ($n = 3$ slices from 5 mice per group).

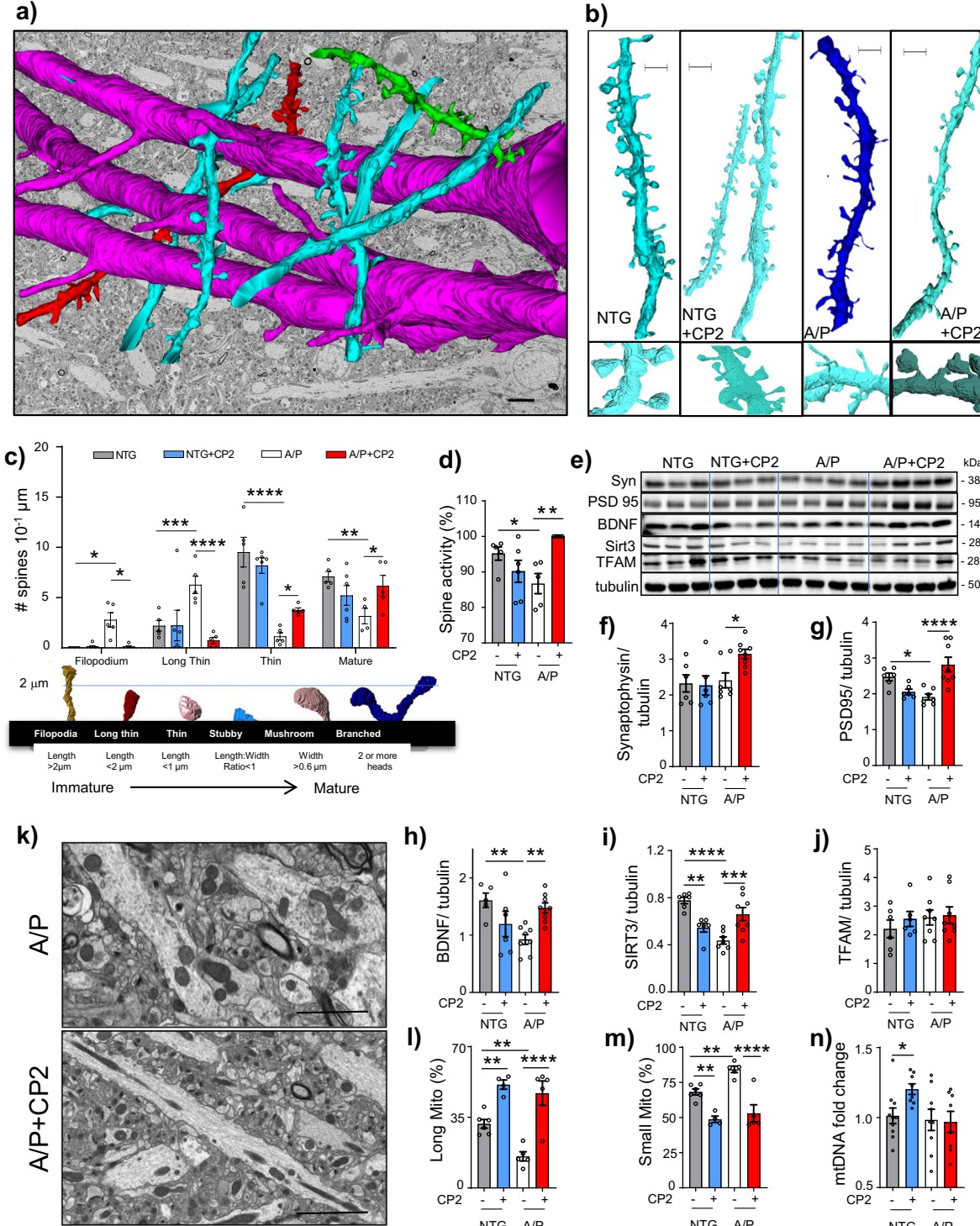

**Fig. 6 CP2 treatment restores dendritic spine morphology and mitochondrial dynamics in the hippocampus of APP/PS1 mice. a** Representative image of 3DEM reconstruction of axons (red) and dendrites (green, blue) from the CA1 hippocampal region of a NTG mouse. Reconstruction is superimposed on 2DEM from the same brain. Scale bar, 5 μm. **b** Representative 3DEM reconstructions of dendrites from CA1 region of NTG and APP/PS1 mice. Scale bar, 1 μm. **c** Quantification of dendritic spine morphology in vehicle and CP2-treated NTG and APP/PS1 mice. Mature dendritic spines included stubby, mushroom and branched. **d** Quantification of active synapses visualized using 3DEM. **e** Western blot analysis in the hippocampal tissue assaying levels of synaptophysin (Syn), BDNF, PSD95, TFAM, and SIRT3. **f–j** Quantification of proteins from **e**. **k** Representative 2DEM micrographs of mitochondria in the hippocampus of CP2-treated and vehicle-treated APP/PS1 mice are used for quantifying mitochondrial morphology. **l**, **m** CP2 treatment increases the number of elongated mitochondria and decreases the number of small organelles. **n** CP2 increased mitochondrial DNA copy number in NTG mice. Data are presented as mean ± S.E.M. A two-way ANOVA with Fisher's LSD post-hoc test was used. $n = 5$ per group. *$P < 0.05$, **$P < 0.01$, ***$P < 0.001$, ****$P < 0.0001$.

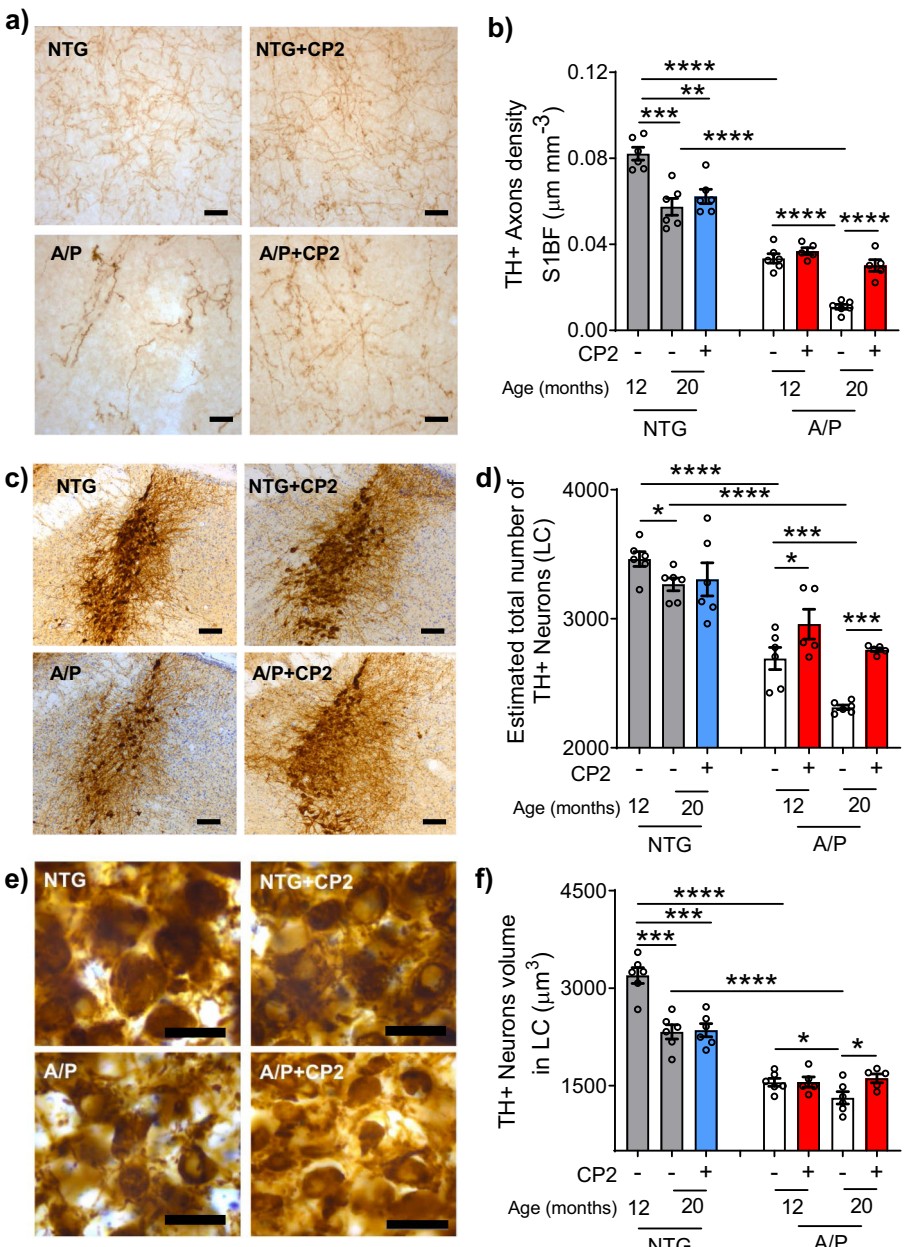

**Fig. 7 CP2 treatment after the onset of AD-like neuropathology halts progressive degeneration of LC neurons in APP/PS1 mice.** APP/PS1 and NTG mice were treated with CP2 or vehicle starting at 10 months of age. The animal brains were harvested at 12 (2 months treatment) or 20 months of age (10 months treatment) and evaluated for the integrity of the NAergic neurotransmitter system. **a** CP2 stops the progressive loss of TH+ axons in the cortex of 20-month-old mice APP/PS1 mice. Representative images of TH+ axonal projections in *S1BF*. Scale bar, 20 μm. **b** The density (μm/mm³) of TH+ axons in *S1BF* was determined using stereological length estimation using spherical probe and images presented in **a**. Compared to NTG mice, APP/PS1 mice exhibit a significant progressive loss of TH+ axons at 12 and 20 months of age. CP2 treatment prevented loss of TH+ axons in APP/PS1 mice occurs between 12 and 20 months of age. **c** Representative image of TH+ LC neurons in 20-month-old mice. Scale bar, 100 μm. **d** CP2 stops the progressive loss of TH+ neurons in APP/PS1 mice in LC. **e** Higher magnification images from **c** were used to evaluate the relative sizes of neurons. Scale bar, 50 μm. **f** CP2 stops the progressive loss of TH+ neuronal volume in APP/PS1 mice at 20 months of age. $n = 6$ female mice per group. Data are presented as mean ± S.E.M. A two-way ANOVA with Fisher's LSD post-hoc test was used to analyze the differences between APP/PS1 mice, and between untreated groups of NTG and APP/PS1 mice. A Student *t*-test was used to analyze the differences between untreated and CP2-treated NTG mice. *$P < 0.05$, **$P < 0.01$, ***$P < 0.001$, ****$P < 0.0001$.

old CP2-treated APP/PS1 mice were comparable to those in 12 month old APP/PS1 mice (Fig. 7b, d, f; 20-month-old group). These data demonstrate that CP2 specifically protects the neuronal network in APP/PS1 mice, which might be associated with the reduction of Aβ accumulation and toxicity.

**CP2 treatment activates translational neuroprotective mechanisms essential for human AD.** To investigate further

mechanisms associated with CP2 efficacy, we performed next-generation RNA sequencing (RNA-seq) using brain tissue from vehicle-treated or CP2-treated NTG and APP/PS1 mice (Fig. 8). Principal component analysis (PCA) revealed a good separation among all groups (Supplementary Fig. 13a). The comparison between vehicle-treated NTG and APP/PS1 mice identified 3320 differentially expressed genes (DEGs) (Fig. 8a, Supplementary Data 2). The top functional changes associated with these DEGs

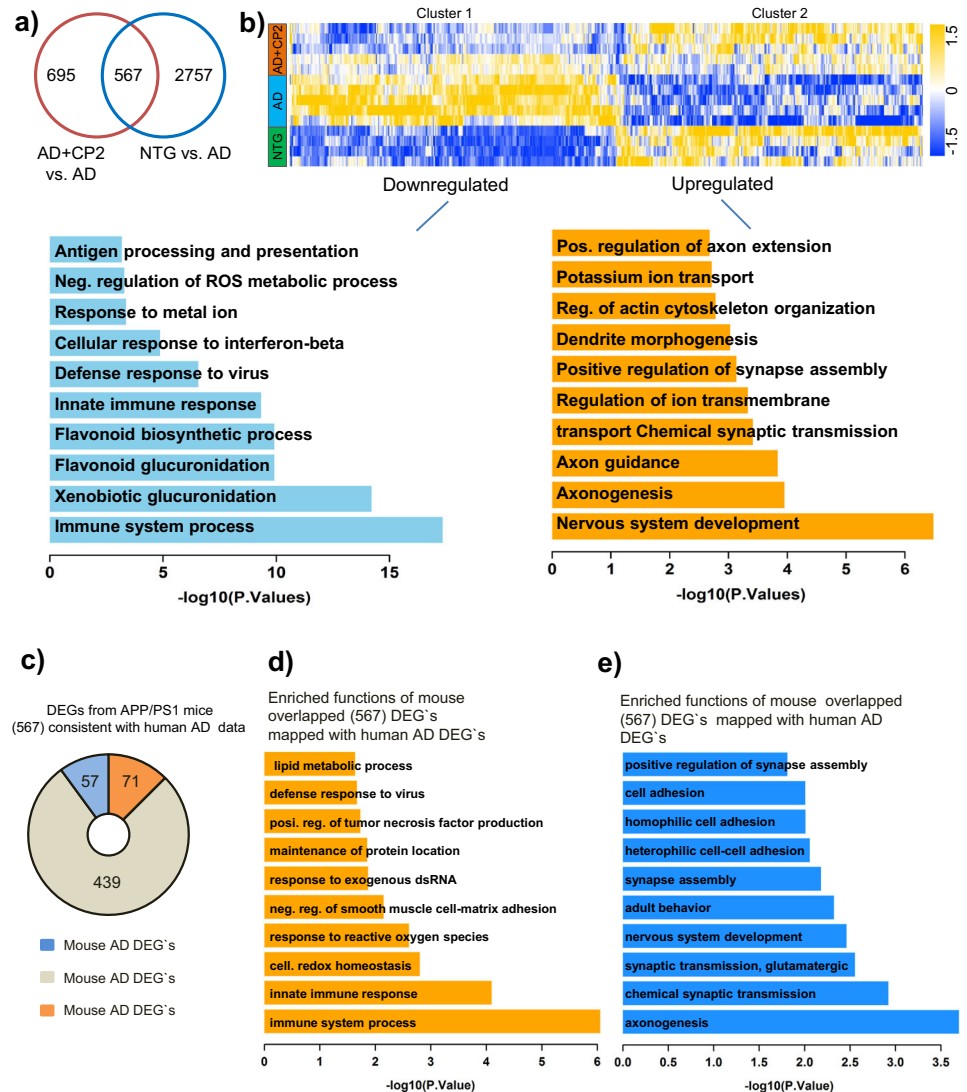

**Fig. 8 Global gene expression patterns in brain tissue of APP/PS1 mice treated with CP2 relative to NTG and APP/PS1 mice. a** Venn diagram of differentially expressed genes ($P < 0.05$) in the brain tissue of vehicle-treated and CP2-treated NTG and APP/PS1 mice. Overlapped DEGs (567) represent specific gene pools affected by CP2 treatment. **b** Heatmap of the overlapped 567 genes shows two clusters where CP2 treatment reversed (upregulated or downregulated) the expression of a subset of genes in APP/PS1 mice to the levels observed in NTG littermates. Gene function enrichment analysis shows pathways associated with downregulated (Cluster 1) or upregulated (Cluster 2) genes after CP2 treatment in APP/PS1 mice. **c** Pie diagrams showing the number of downregulated (upper graph) and upregulated (lower graph) DEGs in vehicle-treated APP/PS1 mice that correlate with corresponding downregulated or upregulated DEGs in the human AMP-AD dataset, respectively. **d** Enriched functions of downregulated DEGs in vehicle-treated APP/PS1 mice that correlate with the identified downregulated DEGs in the human AMP-AD dataset. **e** Enriched functions of upregulated DEGs in vehicle-treated APP/PS1 mice that correlate with the identified upregulated DEGs in the human AMP-AD dataset. All mice were 23-month-old treated with CP2 or vehicle for 13–14 months. NTG, $n = 4$ mice per group; APP/PS1 and APP/PS1 + CP2, $n = 5$ mice per group.

are listed in Supplementary Data 3, 4 and Supplementary Fig. 13b. Processes affected by the disease in APP/PS1 mice overlap with pathways well-established in AD patients including ATP metabolism, ion transport, nervous system development, synaptic transmission, and inflammation[51–56] (Supplementary Fig. 13b). Comparison of CP2-treated and vehicle-treated APP/PS1 mice revealed changes in 1262 DEGs (Fig. 8a, Supplementary Data 5). The top biological processes associated with these DEGs included inflammatory response, redox signaling, nervous system development, and regulation of axonal guidance (Supplementary Fig. 13c–e, Supplementary Data 6, 7). Out of 3320 genes differentially affected in vehicle-treated APP/PS1 vs. NTG mice, the expression of 567 genes was reverted by CP2 treatment to the levels detected in NTG mice (Fig. 8b, Supplementary Data 8). A heatmap of changes in these 567 DEGs shows two clusters with a

subset of genes that were either downregulated or upregulated by CP2 treatment (Fig. 8b, Clusters 1 and 2, Supplementary Data 9, 10). Gene function enrichment analysis showed that pathways downregulated by CP2 in APP/PS1 mice included oxidative stress and immune response (Supplementary Fig. 14d, e), consistent with data generated in the brain and periphery of CP2-treated APP/PS1 mice (Fig. 4).

Among genes involved in the regulation of oxidative stress and apoptosis were *G6pdx*, *BIRC3*, *TRIM30a*, *Trp53*, and *Mt3*, all known to play a role in human disease (Supplementary Fig. 14e). Other important changes included global downregulation of the immune response by CP2 including the acute phase response, such as upregulation of *SERPING1*, interferon signaling (*DDX58*, *FCGR1A*, *TRIM25*, *GBP5*, *TLR7*, *IFITM3*, *IFIT1*, *NLRC5*, *OASL2*, *IRGM1*, *IIGP1*), and major histocompatibility complex (MHC)

class II presentation (*H2-DMa, H2-Q10, H2-Eb1, H2-Aa, PSMB8, PSMB9*) (Supplementary Fig. 14e). Pathways upregulated by CP2 included dendritic spine maturation, axonal extension and guidance, and synaptic transmission (Supplementary Fig. 14f-g). The identified upregulated genes in dendrite morphogenesis pathways included the Down syndrome cell-adhesion molecule (*DSCAM*) (Supplementary Fig. 14f), which is involved in governing neurite arborization, mosaic tiling, and dendrite self-avoidance and BTB Domain Containing 3 (*BTBD3*), which has a role in dendritic guidance toward active axon terminals. Genes that mediate axonogenesis, including *Ntng1* and *Ntng2*, and axonal guidance, were also upregulated in CP2-treated APP/PS1 mice (Supplementary Fig. 14g, Supplementary Fig. 13d). Additional genes upregulated by CP2 in APP/PS1 mice included those involved in synaptic transmission and synapse assembly and that are known to be downregulated in AD patients, including glutamate receptor 4 (*GRIA4*), which mediates fast synaptic excitatory neurotransmission; metabotropic glutamate receptor 7 and 2 (*GRMN7* and *GRM2*), which facilitates the formation of LTP; Double C2 protein (*Doc2a*), which contributes to spontaneous excitatory and inhibitory release; and Neurexins1 (*NRXN1*), which facilitates formation of functional synaptic structures (Supplementary Figs. 13e, 14h). These data are consistent with the improved synaptic function in CP2-treated APP/PS1 mice (Fig. 5).

To provide further evidence for the translational potential of our findings, we cross-validated transcriptomic data from our study with the human brain transcriptome available through coexpression meta-analysis across the Accelerating Medicines Partnership in Alzheimer's Disease Target Discovery and Preclinical Validation Project (AMP-AD–ampadportal.org)[57]. RNA-seq AMP-AD data were generated across three large scale but distinct human *postmortem* brain studies collected from 2114 samples across 7 brain regions and 3 research studies[54–56]. Since in our study we used only female mice, we restricted human data to females only. We first correlated genetic changes found in our comparison of NTG vs. APP/PS1 mice to significant DEGs identified in comparison between the AD vs. control female cohort in the AMP-AD set. Out of 3114 downregulated DEGs in the human AD cohort (Supplementary Fig. 14a, Supplementary Data 11), we identified 294 mouse DEGs that matched the human gene set (Supplementary Data 12). Functional enrichment analysis showed that the most downregulated pathways in both human and mouse AD were involved in synaptic transmission, nervous system development, histone deacetylation, and axonogenesis (Supplementary Fig. 14b, Supplementary Data 13). Downregulated shared genes in mouse and human AD included genes involved in pyruvate metabolism such as *MCP2, DLAT*, and *PDHB* (Supplementary Data 13). Among 518 upregulated DEGs shared between human and mouse AD, top functions were enriched for the innate immune response (Supplementary Fig. 14a, c, Supplementary Data 14, 15). Thus, APP/PS1 mice recapitulate major pathways affected in human AD.

We next compared the 567 DEGs associated with CP2 treatment in APP/PS1 mice (Fig. 8a) with DEGs from females in the AMP-AD RNA-seq data collection. We found that 128 out of the total 567 overlapping mouse AD DEGs corresponded to human AD genes (Fig. 8c). CP2 treatment in APP/PS1 mice reversed the expression of 71 genes that were upregulated in both mouse and human AD (Supplementary Data 16). Functional enrichment analysis showed that these 71 genes were involved in the regulation of the immune processes, inflammation, response to reactive oxygen species, and TNF production (Fig. 8d, Supplementary Data 17). In contrast, CP2 reversed the expression of 57 deregulated genes that are involved in axonogenesis, glutamatergic synaptic transmission, nervous system

development, and synapse assembly (Fig. 8c, e and Supplementary Data 18, 19). Taken together, these data demonstrate that AD-associated transcriptional and functional changes observed in our mouse model of AD are counteracted by CP2 treatment. CP2-treated APP/PS1 mice showed attenuated expression of a number of genes involved in neuroinflammatory processes, consistent with our observation of a decreased number of activated astrocytes and microglia. Moreover, CP2 treatment of APP/PS1 mice restored expression of genes involved in neurotransmission, dendritic morphology and axonal guidance and extension, which were linked to the restoration of hippocampal LTP and increased number of mature dendritic spines (Fig. 6). These data demonstrate that pathways improved by CP2 treatment in APP/PS1 mice comprise major pathways essential for therapeutic efficacy in AD patients.

## Discussion

AD is associated with early energy hypometabolism, synaptic and mitochondrial dysfunction, oxidative stress, inflammation, abnormal proteostasis and progressive neurodegeneration. Here, we demonstrate that mild energetic stress associated with partial inhibition of MCI induces activation of integrated stress-response mechanisms that attenuate effects of pathological pathways such as abnormal energy homeostasis, synaptic dysfunction, and inflammation, ultimately blocking neurodegeneration in a translational mouse model of AD, the APP/PS1 mice. The therapeutic efficacy achieved has translational relevance, as the intervention was started after the onset of Aβ neuropathology[26], cognitive symptoms[27], bioenergetic dysfunction[28], and progressive neurodegeneration. Beneficial mechanisms affected by CP2 treatment in APP/PS1 mice overlap with signatures established in AD patients, females in particular, supporting the high translational potential of this approach. Major translational targets affected by CP2 treatment included the immune system response and multiple pathways involved in synaptic function and neurotransmission, which underlie early pathology in AD patients[8]. Since CP2 improved axonogenesis and dendritic spine morphology and function, it is feasible that this treatment could also induce neuronal regeneration.

The strength of our study is in the utilization of multiple early and late disease outcome measures in a mouse model of AD that closely mimics neuropathological mechanisms of human disease. In particular, we hypothesized that the general failure of the preclinical studies in mouse models of AD to predict outcomes of human clinical trials is related to the reliance on treatments of younger mice and a very limited set of neuropathological outcome measures. Thus, it is important that CP2 treatment of APP/PS1 mice was conducted when AD-like pathology, including progressive neurodegeneration, was well established and the broad array of measures, including advanced imaging techniques and translational biomarkers, were applied in vivo and in tissue, further supporting the ability to monitor therapeutic efficacy of this approach in humans. However, this study was limited only to female mice.

While the details and the hierarchy of molecular mechanisms involved in neuroprotective stress response require further evaluation, AMPK activation appears to play a central role. Indirect activation of AMPK by inhibition of MCI has been shown to increase life span, rejuvenate the transcriptome, and protect from neurodegeneration[58–62]. Paradoxically, these manipulations improved MCI assembly, increased complex I-linked state 3 respiration, and decreased ROS production[59]. Studies in a cohort of 2200 ultranonagenarians revealed that mutations in subunits of MCI that resulted in its partial inhibition had beneficial effect on longevity[63]. The compelling support for safety of the application

of MCI inhibitors in humans comes from metformin, an FDA approved drug to treat diabetes. Among other targets, metformin inhibits MCI. It is prescribed to the elderly population and has relatively safe profile even after chronic treatment[64]. Recent study conducted in a large Finish population of older people with diabetes demonstrated that long-term and high-dose metformin use does not increase incidences of AD and is associated with a lower risk of developing AD[65]. Resveratrol is another MCI modulator where its effect on MCI (activation or inhibition) depends on the concentration. It also inhibits mitochondrial Complex V. Resveratrol is currently in clinical trials for multiple human conditions[66–68]. Compared to CP2, these compounds have limitations associated with the lack of selectivity, specificity, and bioavailability.

CP2 treatment effectively reduced Aβ accumulation. This might be explained by an additional ability of CP2 to bind Aβ peptides reducing their toxicity, which together with the AMPK-dependent mechanisms, could produce a synergistic effect[69]. Our work also provides evidence that CP2 treatment improves mitochondrial dynamics and function. While counterintuitive, given that progressive mitochondrial dysfunction is well characterized in AD patients and APP/PS1 mice, mild MCI inhibition in dysfunctional mitochondria could decrease ROS production, while in functional mitochondria could improve energetics, as we demonstrated previously[69], and through activation of stress responses, could promote biogenesis and mitophagy, contributing to a healthier mitochondrial pool and more effective energy production. Positive effects on mitochondrial function and neuroprotection are further supported by an increase in Sirt3 levels. In AD patients and AD models, reduction of Sirt3 was directly linked to the loss of synaptic function, Aβ and Tau pathology, and neurodegeneration[20,21,44]. While APP/PS1 mice utilized in our study do not have pTau accumulation, increased levels of Sirt3 and decreased activity of GSK3β suggest that CP2 treatment could also be effective in reducing Tau toxicity. Indeed, our independent study in the mouse model of AD that along with the mutant human APP and PS1 proteins also expresses mutant human Tau protein, the 3xTg-AD mice[70], demonstrated that chronic CP2 treatment reduced pTau levels and improved LTP and energy homeostasis in symptomatic male and female mice. While this study interrogated CP2 efficacy only in female APP/PS1 mice, treatment in 3xTg-AD mice was beneficial in males and females. These data further support the concept that targeting mitochondria with small molecule specific MCI inhibitors represents a promising strategy that could be efficacious in patients with synergistic Aβ-related and pTau-related pathology. While not as pronounced as in APP/PS1 mice, CP2 treatment improved health parameters in aged NTG mice, reducing oxidative stress and cellular senescence, key pathways that accelerate aging, and improving body mass, glucose tolerance, mitochondrial function, and physical strength. Taken together, the results of our studies suggest that partial inhibition of MCI represents a novel treatment strategy for blocking neurodegeneration and cognitive impairment, even after the development of AD-like symptoms, and also could enhance healthspan delaying the onset of multiple age-related diseases, including AD.

## Methods

All experiments with mice were approved by the Mayo Clinic Institutional Animal Care and Use Committee in accordance with the National Institutes of Health's Guide for the Care and Use of Laboratory Animals. IACUC Protocol A00001186-16-R18.

**CP2 synthesis**. CP2 was synthesized by the Nanosyn, Inc biotech company (http://www.nanosyn.com) as described[71] and purified using HPLC. Authentication was through NMR spectra to ensure the lack of batch-to-batch variation in purity. CP2

was synthesized as free base. For in vitro experiments, CP2 was prepared as 50 mM stock solution in PEG400. Stock aliquots of 20 μl were stored at −80 °C. Each aliquot was used once to avoid freeze-though cycle. Preparation of CP2 for mouse treatment via drinking water is described below.

**Mice**. The following female mice were used in the study: double transgenic APP/PS1[72] and their non-transgenic (NTG) littermates; CD-1 and C57BL/6J wild-type mice. Genotypes were determined by PCR as described in ref. [72]. All animals were kept on a 12 h–12 h light–dark cycle, with a regular feeding and cage-cleaning schedule. Mice were randomly selected to study groups based on their age and genotype. Mice were housed 5 per cage, water consumption and weight were monitored weekly. CP2 concentration was adjusted based on mouse weight/water consumption weekly. The number of mice in each group was determined based on the 95% of chance to detect changes in 30–50% of animals. The following exclusion criteria were established: significant (15%) weight loss, changing in the grooming habits (hair loss), pronounced motor dysfunction (paralyses), or other visible signs of distress (unhealed wounds).

**Human brain tissue**. Experiments with post-mortem human brain tissue were approved by the Mayo Clinic IRB (#12-007847) and were carried out in accordance with the approved guidelines. Informed consent was obtained from all subjects involved in the study. One brain specimen with a postmortem interval of 6 h from a cognitively normal female 99 years old was obtained from Mayo Clinic. Cortex was used for mitochondrial isolation to determine CP2-dependent inhibition of MCI.

**Mitochondrial isolation and measurements of ETC complex activity**. Intact brain mitochondria were isolated from post-mortem human or mouse brain tissue using differential centrifugation with digitonin treatment[73]. Brain tissue was immersed into ice-cold isolation medium (225 mM mannitol, 75 mM sucrose, 20 mM HEPES-Tris, 1 mM EGTA, pH 7.4), supplemented with 1 mg/ml BSA. Tissue was homogenized with 40 strokes by pestle "B" (tight) of a Dounce homogenizer in 10 ml of isolation medium, diluted two-fold, and transferred into centrifuge tubes. The homogenate was centrifuged at 5900 × g for 4 min in a refrigerated (4 °C) Beckman centrifuge. The supernatant was centrifuged at 12,000 × g for 10 min and pellets were resuspended in the same buffer, and 0.02% digitonin was added. The suspension was homogenized briefly with five strokes in a loosely fitted Potter homogenizer and centrifuged again at 12,000 × g for 10 min, then gently resuspended in the isolation buffer without BSA and washed once by centrifuging at 12,000 × g for 10 min. The final mitochondrial pellet was resuspended in 0.1 ml of washing buffer and stored on ice. The respiratory activities were measured in Oroboros high-resolution respirometer as previously described[74].

The activity of the ETC complexes was measured spectrophotometrically using a plate reader (SpectraMax M5, Molecular Devices, USA) in 0.2 ml of standard respiration buffer composed of 125 mM sucrose, 25 mM Tris-HCl (pH = 7.5), 0.01 mM EGTA, and 20 mM KCl at 25 °C. NADH-dependent activity of complex I was assayed as oxidation of 0.15 mM NADH at 340 nm (ε340 nm = 6.22 mM$^{-1}$cm$^{-1}$) in the assay buffer supplemented with 10 μM cytochrome $c$, 40 μg/ml alamethicin, 1 mM MgCl$_2$ (NADH media). NADH:Q reductase was measured in NADH media containing 2 mg/ml BSA, 60 μM decylubiquinone, 1 mM cyanide and 5–15 μg protein per well. Only the rotenone (1 μM)-sensitive part of the activity was used for calculations. NADH:HAR reductase was assayed in NADH media containing 1 mM HAR and 2–5 μg protein per well. Complex II succinate:DCIP reductase activity was recorded at 600 nm (ε600 nm = 21 mM$^{-1}$cm$^{-1}$) in the KCl assay buffer (125 mM KCl, 20 mM HEPES-Tris, 0.02 mM EGTA, pH 7.6) containing 15 mM succinate, 40 μM decylubiquinone, 0.1 mM DCIP, 1 mM KCN, and 5–10 μg protein per well. Complex IV ferrocytochrome $c$ oxidase activity was measured as oxidation of 50 μM ferrocytochrome $c$ at 550 nm (ε550 nm = 21.5 L·mM$^{-1}$cm$^{-1}$) in KCl assay buffer supplemented with 0.025% dodecylmaltoside and 1–3 μg protein per well. To assess the effect of CP2 on the activity, we pre-incubated 20–40 μg/ml mitochondria with various concentrations of CP2 for 10 min at 25 °C in the absence of substrates and then measured the residual activity as described above.

**In vitro pharmacology studies**. CP2 binding and enzyme and uptake assays were conducted by the Contract Research Organization (CRO) Eurofins Cerep (France). CP2 was tested at 10 μM. Compound binding was calculated as a % inhibition of the binding of a radioactively labeled ligand specific for each target. Compound enzyme inhibition effect was calculated as a % inhibition of control enzyme activity. In each experiment and if applicable, the respective reference compound was tested concurrently with the test compounds, and the data were compared with historical values determined at Eurofins. The experiment was accepted in accordance with Eurofins validation Standard Operating Procedure. Results showing an inhibition (or stimulation for assays run in basal conditions) higher than 50% are considered to represent significant effects of the test compounds. 50% is the most common cut-off value for further investigation (determination of IC50 or EC50 values from concentration-response curves). Results showing an inhibition (or stimulation) between 25 and 50% are indicative of weak to moderate effects (in most assays, they should be confirmed by further testing as they are within a range

where more inter-experimental variability can occur). Results showing an inhibition (or stimulation) lower than 25% are not considered significant and mostly attributable to variability of the signal around the control level. Low to moderate negative values have no real meaning and are attributable to variability of the signal around the control level. High negative values (≥50%) that are sometimes obtained with high concentrations of test compounds are generally attributable to non-specific effects of the test compounds in the assays. On rare occasion they could suggest an allosteric effect of the test compound. *Binding Assays.* The results are expressed as a percent of (control specific binding/measured specific binding) × 100 control specific binding and as a percent inhibition of control specific binding (100 − (measured specific binding/control specific binding) × 100 obtained in the presence of the test compounds. The IC50 values (concentration causing a half-maximal inhibition of control specific binding) and Hill coefficients (nH) were determined by non-linear regression analysis of the competition curves generated with mean replicate values using Hill equation curve fitting $Y = D + [(A - D/1 + (C/C50)^{nH}]$ where $Y$ = specific binding, $A$ = left asymptote of the curve, $D$ = right asymptote of the curve, $C$ = compound concentration, C50 = IC50, and nH = slope factor. This analysis was performed using software developed at Cerep (Hill software) and validated by comparison with data generated by the commercial software SigmaPlot® 4.0 for Windows® (© 1997 by SPSS Inc.). The inhibition constants (Ki) were calculated using the Cheng Prusoff equation $K_i = (IC_{50}/(1 + L/K_D)$ where $L$ = concentration of radioligand in the assay, and $K_D$ = affinity of the radioligand for the receptor. A scatchard plot is used to determine the $K_D$. *Enzyme and Uptake Assays.* The results are expressed as a percent of control specific activity: (measured specific activity/control specific activity) × 100 and as a percent inhibition of control specific activity 100-(measured specific activity/control specific activity × 100) obtained in the presence of the test compounds. This analysis was performed using software developed at Cerep (Hill software) and validated by comparison with data generated by the commercial software SigmaPlot® 4.0 for Windows® (© 1997 by SPSS Inc.).

**250 kinase panel kinase panel**. Nanosyn, Inc (Santa Barbara, CA, http://www.nanosyn.com.) was contracted to conduct Kinome Wide panel (KWP) screening against 250 kinases. CP2 was tested at 1 and 10 µM concentrations. The buffer components and assay conditions differ based on the specific assay. The assay has a total volume of 10 µL comprised of 5 µL enzyme buffer and 5 µL substrate buffer. After incubation and termination, substrate and product are separated and quantified electrophoretically using the microfluidic-based LabChip 3000 Drug Discovery System from Caliper Life Sciences. ADP Glo is another method of detection used in some assays where data is obtained by quantifying the intensity of luminescence.

*Summary of assay.*

1. Compound management
2. Assay setup
3. Reaction termination and data generation

*Compound management.* Compounds are plated at 100× of the screening concentrations on a Labcyte compatible plate along with 3 reference compounds.

*Assay setup.* Add 5 µl/well of enzyme buffer to the 384-well assay plate except control wells where buffer without enzyme are added. Using Labcyte Echo 550, transfer 100 nl of compounds to the assay plate. Add 5 µl/well of substrate buffer to the assay plate. Incubate the reaction mixture at 25 °C according to the assay specific incubation time.

*Reaction termination and data generation.* For Mobility Shift assay, the reaction is terminated by addition of EDTA buffer and the assay plate is read on Caliper LabChip 3000.

For ADP Glo based assay, the reaction is terminated by applying the Promega ADP Glo Kinase Assay kit and the assay plate is read on a LJL Biosystems Analyst HT.

**In vivo CP2 pharmacokinetic**. CP2 bioavailability was determined using C57BL/6J female mice ordered from the Jackson Laboratory. Mice were acclimatized for one week to the new environment prior to initiation of experiments. For evaluating CP2 concentrations in plasma, mice were injected with a single intravenous (IV) dose of CP2 (3 mg/kg in DMSO) in a lateral tail vein using a 0.5 cc tuberculin syringe. An independent cohort of mice was administered CP2 by oral gavage (PO) (25 mg/kg in 20% PEG 400 and 5% dextrose water in PEG) with a 1 cc tuberculin syringe with a stainless steel 22 gauge straight feeding needle. At 0, 0.08, 0.25, 0.5, 1, 2, 4, 8, and 24 h after treatment, mice were anesthetized, and 200 µl of blood was collected from the retro-orbital sinus through a K2EDTA coated capillary into a K2EDTA coated microtainer tube. Plasma was separated by centrifugation at 4 °C (10,000 rpm × 3 min), transferred to a microcentrifuge tube, immediately frozen on dry ice, and stored at −80 °C until the analysis. Pharmacokinetic parameters for CP2 were estimated with standard non-compartmental analysis. Pharmacokinetic parameters for plasma were estimated using the concentration–time profiles for

each route of administration, IV or PO (Supplementary Fig. 4). The apparent elimination half-life (T1/2) was calculated using a pharmacokinetics half-life program on a RS/1 computer system. The area under the plasma concentration–time curve (AUC) from time 0 to infinite time (AUC (0-inf)) was determined by conventional trapezoidal summation and extrapolation. The maximum plasma CP2 concentration (C_max) and the time of maximum plasma CP2 concentration (Tmax) were read directly from the plotted data. The AUC (0–24 h) represents the area under the plasma concentration-time curve over the last 24 h dosing interval; the AUC (0-inf), area under the plasma concentration-time curve to infinity represents the total drug exposure across time. CI/F, apparent total clearance of the drug from plasma after oral administration and F represents the drug bioavailability. The PO availability of CP2 is 65%.

**CP2 quantification using LC-MS/MS**. Paclitaxel (Sigma, St. Louis, MO) was used as an internal standard (IS). Ultra-pure water was generated using a Barnstead nanopure diamond system (Thermo, Marietta, OH). Optima LC/MS-grade methanol (MeOH, Fisher Scientific, Waltham, MA), analytical grade formic acid (Fisher Scientific, Waltham, MA), 96 well protein crash plates and 1 ml 96 well polypropylene collection plates (Chromtech, Apple Valley, MN), Kunststoff-Kapillaren end-to-end K2EDTA coated plastic 30 µl capillary tubes (Fisher Scientific, Waltham, MA) and K2EDTA presprayedmicrotainer collection tubes (500 µl, Becton, Dickinson and Company, Franklin Lakes, NJ) were utilized in this study. For controls, drug-free mouse plasma was obtained from healthy CD-1 mice containing 0.1% K2EDTA that was purchased from Valley Biomedical and stored at −20 °C for later use. The LC-MS/MS consisted of a Waters Acquity H class ultra-performance liquid chromatography (UPLC) system containing a quaternary solvent manager and sample manager-FTN coupled to a Xevo TQ-S mass spectrometer equipped with an electrospray ionization (ESI) source. Data were acquired and analyzed using Waters MassLynx v4.1 software. Detection of CP2 and paclitaxel (internal standard, IS) was accomplished by multiple reaction monitoring (MRM) using the mass spectrometer in positive ESI mode with capillary voltage, 2.5 kV; source temperature 150 °C; desolvation temperature 400 °C; cone gas flow 150 L/h; desolvation gas flow 800 L/h. The cone voltages and collision energies for CP2 and paclitaxel were determined by MassLynx-Intellistart v4.1 software and were 12 and 18 V (cone) and 30 and 66 eV (collision), respectively. MRM precursor and product ions were monitored at m/z 394.41 > 139.19 for CP2 and 854.29 > 105.08 for paclitaxel. Data were collected from 2–5.5 for both CP2 and paclitaxel. The separation of CP2 and paclitaxel was achieved using an Agilent Poroshell 120 EC-C18 (2.7 µ, 2.1 × 100 mm) with an Agilent EC-C18 pre-column (2.7 µ, 2.1 × 5 mm) and a gradient elution program containing ultra-pure water and MeOH, both with 0.1% formic acid. The gradient begins with 70% aqueous, decreases to 10% aqueous over 3 min and holds there for 2 min, then returns to baseline over 0.1 min and holds to equilibrate for 2.9 min. The flow rate was 0.4 ml/min, the total run time was 8 min, injection volume was 5 µl, and the column and auto sampler temperatures were 40 °C and 20 °C, respectively. Stock solutions of CP2 (5 mg/ml, dissolved in DMSO) and paclitaxel (200 ng/ml, dissolved in acetonitrile, ACN) were prepared in salinized glass vials and stored at −20 °C. 20× working stock solutions were prepared daily and diluted in 1:1 MeOH:H2O and stored at −20 °C. Plasma standards containing CP2 (0.2–100 ng/ml) were prepared by adding (5 µl) aliquots of 20× CP2 to plasma (95 µl) in 1.5 ml slick microfuge tubes, 50 µl of this plasma dilution were transferred to a 96 well protein crash plate. Analytes were isolated using protein precipitation with 150 µl ACN containing IS (200 ng/ml). The plate was capped and shaken for 20 min at 1100 rpm. The sample was then vacuum filtered into a Chromtech (Apple Valley, MN) 1 ml 96 well polypropylene collection plate, evaporated to dryness under a gentle stream of nitrogen, and reconstituted with 200 µl H2O/ACN (1:1). The collection plate was capped and shaken at 900 rpm for 20 min, and 5 µl aliquots were injected into the LC/MS/MS. Mass Spectra and ion chromatograms of CP2 and IS were processed using MasSLynx v4.1 software with TargetLynx. Standard curves for CP2 and paclitaxel were analyzed using the peak area ratio of CP2 vs. IS. CP2 standard curve concentrations were 0.2, 0.5, 1, 5, 10, 20, 50, and 100 ng/ml. The concentration of CP2 in the brain was determined using the same approach.

**Acute CP2 treatment in symptomatic APP/PS1 mice**. A cohort of 9–10-month-old female APP/PS1 mice was treated with CP2 (25 mg/kg in 20% PEG 400 and 5% dextrose water in PEG) by oral gavage. Mice were sacrificed at 0, 4, 24, 48, and 72 h (n = 1 per time point), and Western blot analysis was conducted in the hippocampal tissue to determine the engagement of neuroprotective mechanisms. In an independent cohort of 9–10-month-old female and male APP/PS1 mice and their NTG controls (n = 3 mice per group), baseline glucose uptake was measured at 0 h and after CP2 gavage (25 mg/kg in 20% PEG 400 and 5% dextrose water in PEG) at 24, 48, and 72 h using in vivo FDG-PET as discussed below.

**In vivo FDG-PET**. Mice were fasted one hour prior to I. P. injection of 270 uCi of fludeoxyglucose F18 (18FDG) in 200 µl injection volume prepared the same day at the Mayo Clinic Nuclear Medicine Animal Imaging Resource. Imaging was conducted 30 min post injection. Prior to imaging, mice were individually placed in an anesthesia machine (Summit Medical Equipment Company, Bend, Oregon) and anesthetized with 4% isoflurane with 1–2 LPM oxygen. Anesthesia was further

maintained with 2% isoflurane delivered by a nose cone. Mice were placed in MicroPET/CT scanner (Scanner Inveon Multiple Modality PET/CT scanner, Siemens Medical Solutions USA, Inc.). Mouse shoulders were positioned in the center of the field of view (FOV), and PET acquisition was performed for 10 min. CT scanning parameters were as following: 360 degree rotation; 180 projections; Medium magnification; Bin 4; Effective pixel size 68.57; Tranaxial FOV 68 mm; Axial FOV 68 mm; 1 bed position; Voltage 80 keV; Current 500 µA; Exposure 210 ms. CT reconstruction parameters: Alogorithm: Feldkamp, Downsample 2, Slight noise reduction with application of Shepp-Logan filter. A final analysis was done using PMOD Biomedical Image Quantification and Kinetic Modeling Software, (PMOD Technologies, Switzerland). The volume of interest (VOI) was created on the CT image of the entire brain. The VOI was applied to the corresponding registered PET scan. The volume statistics were recorded. The percentage of brain glucose uptake was calculated by correcting the measured concentration of uCi recorded after 30 min to the amount of injected dose.

**Chronic CP2 treatment in NTG and symptomatic APP/PS1 mice**. NTG and APP/PS1 female mice ($n = 16$–21 per group) were given either CP2 (25 mg/kg/day in 0.1% PEG dissolved in drinking water ad lib) or vehicle-containing water (0.1% PEG) starting at 9 months of age as we described in ref. [14]. Mice were housed 5 per cage, water consumption and weight were monitored weekly. CP2 concentration was adjusted based on mouse weight/water consumption weekly. Independent groups of mice were continuously treated for 14 months until the age of 23 months. Seven to eight months after CP2 treatment, mice were subjected to the battery of behavior tests, metabolic cages (CLAMS), $^{31}$P NMR spectroscopy, and electrophysiology. After mice were sacrificed, tissue and blood were subjected to Western blot analysis, profiling for cytokines/chemokines, next-generation RNA sequencing, immunohistochemistry, electron microscopy examination, and metabolomics as described below.

**Behavior battery**. Behavioral tests were carried out in the light phase of the circadian cycle with at least 24 h between each assessment as we described previously[13]. More than one paradigm ran within 1 week. However, no more than two separate tests were run on the same day. Behavioral and metabolic tests were performed in the order described in the experimental timeline. *Open field test.* Spontaneous locomotor activity was measured in brightly lit (500 lux) Plexiglas chambers (41 cm × 41 cm) that automatically recorded the activity by photo beam breaks (Med Associates, Lafayette, IN). The chambers were located in sound-attenuating cubicles and were equipped with two sets of 16 pulse-modulated infrared photo beams to automatically record *X*–*Y* ambulatory movements at a 100 ms resolution. Data was collected over a 100-min trial at 30-s intervals. *Hanging bar.* Balance and general motor function were assessed using the hanging bar. Mice were lowered onto a parallel rod ($D < 0.25$ cm) placed 30 cm above a padded surface. Mice were allowed to grab the rod with their forelimbs, after which they were released and scored for success (pass or failure) in holding onto the bar for 30 s. Mice were allowed three attempts to pass the test. Any one successful attempt was scored as a pass. The final score was presented as an average of three trials per animal. *Rotarod test.* The accelerating rotarod (UgoBasile, Varese, Italy) was used to test balance and coordination. It comprised of a rotating drum that accelerated from 5 to 40 rpm over a 5-min periods. The latency of each animal to fall was recorded and averaged across three consecutive trials. *Novel Object Recognition test (NOR)* was used to estimate memory deficit. All trials were conducted in an isolated room with dim light in Plexiglas boxes (40 cm × 30 cm). A mouse was placed in a box for 5 min for acclimatization. Thereafter, a mouse was removed, and two similar objects were placed in the box. Objects with various geometric shape and color were used in the study. Mice were returned to the box, and the number of interrogations of each object was automatically recorded by a camera placed above the box for 10 min. Mice were removed from the box for 5 min, and one familiar object was replaced with a novel object. Mice were returned to the box, and the number of interrogations of novel and familiar objects was recorded for 10 min. Experiments were analyzed using NoldusEthoVision software. The number of interrogations of the novel object was divided by the number of investigations of the familiar object to generate a discrimination index. Intact recognition memory produces a discrimination index of 1 for the training session and a discrimination index greater than 1 for the test session, consistent with greater interrogation of the novel object. *Morris Water Maze (MWM).* Spatial learning and memory were investigated by measuring the time it took each mouse to locate a platform in opaque water identified with a visual cue above the platform. The path taken to the platform was recorded with a camera attached above the pool. Each mouse was trained to find the platform during four training sessions per day for three consecutive days. For each training session, each mouse was placed in the water facing away from the platform and allowed to swim for up to 60 s to find the platform. Each training session started with placing a mouse in a different quadrant of the tank. If the mouse found the platform before the 60 s have passed, the mouse was left on the platform for 30 s before being returned to its cage. If the animal had not found the platform within the 60 s, the mouse was manually placed on the platform and left there for 30 s before being returned to its cage. A day of rest followed the day of formal testing.

**Intra-peritoneal (IP) glucose tolerance test (IPGTT)**. The glucose tolerance test measures glucose clearance after it was delivered using IP injection. Mice were fasted for approximately 16 h, and fasted blood glucose levels were determined before a solution of glucose was administered by IP injection. Subsequently, the blood glucose level was measured at different time points during the following 2 h (0, 20, 30, 60, and 120 min after the injection). Mice were injected with 20% glucose solution based on the body weight (2 g of glucose/kg body weight).

**Intra-peritoneal (IP) insulin sensitivity test (IPIST)**. The insulin sensitivity test measures glucose clearance after insulin is delivered using IP injection. Mice were fasted for approximately 5 h, and fasted blood glucose levels were determined before a solution of insulin (diluted in saline) was administered by IP injection. Subsequently, the blood glucose level was measured at different time points during the following 2 h (0, 20, 30 60, 90, and 120 min after the injection). Mice were injected with 0.5 U/ml insulin solution based on the body weight (0.5 U/kg).

**Aβ ELISA**. Levels of Aβ40 and Aβ42 were determined in brain tissue from 23-month-old APP/PS1 mice treated for 14 months with CP2 ($n = 9$) or vehicle ($n = 9$). Differential fractionation was achieved by collecting fractions with most to least soluble Aβ40 and Aβ42 from brain tissue sequentially homogenized in Tris-buffered saline (TBS, most soluble); in TBS containing 1% Triton X-100 (TBS-TX), and 5 M guanidine in 50 mM Tris-HCl (least soluble), pH 8.0 as described in ref. [13]. Levels of human Aβ40 and Aβ42 were determined by ELISA using antibodies produced in-house as previously published[75].

**Immunohistochemistry**. We followed a protocol described previously[50]. Briefly, for histological analysis, mice were perfused intracardially with 4% paraformaldehyde. Brains were cryoprotected, and serial frozen coronal sections (40 µm) were serially distributed into individual wells of 12-well plates. To facilitate the identification of regions of interest for the quantitative stereological analysis, every 24th section through the entire brain was Nissl stained and compared with the stereotaxic coordinates of the mouse brain[76]. To detect antigens of interest, the sections were incubated in primary antibodies followed by the ABC method (Vector Laboratories) using 3,3'-diaminobenzidine (DAB, Sigma Alrich) as the chromogen for visualization. Antigen retrieval was performed using a Rodent Decloaker for all samples, with an additional 88% formic acid pre-treatment for 4G8-incubated samples. Primary antibodies used are: tyrosine hydroxylase (TH) antibody (rabbit polyclonal, Millipore) for noradrenergic (NAergic) neurites/axons; 4G8 anti-Aβ mouse monoclonal antibody (Biolegend) for amyloid deposits; anti-GFAP rabbit polyclonal antibody (Dako) for astrocytes; and anti-Iba1 rabbit polyclonal antibody (Wako) for microglia. NA and dopaminergic (DA) fibers/neurons were visualized using an anti-tyrosine hydroxylase (TH) antibody (Novus Biologicals). Cresyl violet (CV) was used to stain for nuclei of non-MAergic neurons for neuronal counts.

**Stereological analysis of Aβ deposition and glial reaction**. All stereological analysis was performed using the StereoInvestigator software (MicroBrightField, Colchester, VT)[49,50]. Extent of brain area covered by amyloid deposits (4G8 immunostained area), astrocytes (GFAP), or microglia (Iba1) was measured within the regions of interests (ROI) using the area fraction analysis[49]. The ROIs traced using *The Mouse Brain in Stereotaxic Coordinates*[77] included the barrel field region of primary somatosensory cortex (S1BF; sections between bregma −0.10 to −1.22 mm, posterior to the anterior commissure and anterior to hippocampus) and dorsal hippocampus (dentate, CA1, CA2/3; sections between bregma −1.46 to −2.18 mm); the S1 barrel cortex (S1BF) and dorsal hippocampal regions. For each mouse, every 12th coronal brain sections (4–6 sections) containing the ROIs were immunostained. The sections were analyzed using Stereo Investigator (MBF Bioscience, Williston, VT), with a ×40 objective, where the ROIs were outlined, and the Area Fraction Fractionator probe was used to systematically and randomly allocate sampling sites 400 µm apart in the cortex, and 200 µm apart in the hippocampus. At each sampling site, a 100 × 80 µm counting frame was superimposed, containing markers equally spaced from one another at a distance of 15 µm. The area fraction markers that overlap with immunoreactive tissue were labeled as positive, whereas remaining markers were labeled negative. Because the ratio of untainted and stained markers is proportional to the areas occupied, we calculated the percent of total area that was immunostained. Representative images are included in the paper. The actual immunostained brain sections for additional analysis or examination could be available upon request directed to Dr. M. Lee.

**Stereological analysis of NAergic afferents and neurons**. To determine the length of NAergic axons, we used stereological length estimation with spherical probes (Stereo Investigator; MicroBrightField)[50]. Because the densities of NAergic afferents show substantial regional variation, we focused our analysis on the selected subregions or ROIs (S1BF and dorsal hippocampus) for NAergic afferents. Every eighth section, starting from a start of the ROI regions through the entire ROI region of interest, slices were processed for immunocytochemistry. To determine the axon length, virtual spherical probes were placed within a 40 µm thick section. At each focal plane, concentric circles of progressively increasing and decreasing diameters were superimposed, and the intersections with the

immunoreactive fibers and circles were counted ($Q$). To minimize surface artifacts, a guard volume of 1 µm was used. This method allows for the simple determination of the total length density ($L_V$) and the total length ($L$). To reduce the effects of variations in the area selection, $L_V$ was routinely used for comparison between groups. The axon lengths were measured at 50 random locations through the reference space.

To determine the total number of NAergic neurons in the LC, we used the optical fractionator[78]. Every 4th section of the entire region containing the LC was immunostained for TH and counterstained with CV. Total TH+ and TH− neuron numbers were estimated using the optical fractionator probe; the LC region of interest was traced and magnified using the ×100 objective, and TH+ neurons and TH− nuclei were counted using the counting frame of $40 \times 30$ µm a 1 µm guard, a $130 \times 130$ µm sampling grid, and a dissector height of 10 µm.

For unbiased stereological analysis of neuronal size (area and volume), we used the nucleator feature of the Stereology software[50,78]. Neurons were randomly sampled within the sections used for neuron counts and, using the nucleator feature of the software, we determined neuron volumes by placing four rays through the cell, with the nucleolus serving as the midpoint. The cells were measured if their nuclear membrane intersected or touched the inclusion (green) line. The cells were excluded if the nuclear membrane intersected or touched the exclusion (red) line. The measuring rays were "cut" at the intersection with the cell membrane. If a ray extended into a dendrite, the ray was "cut" at the base of the dendrite. Representative images are included in the paper. Representative images are included in the paper. The actual stained brain sections for additional analysis or examination could be available upon request directed to Dr. M. Lee.

**Inflammatory markers**. After 16 h of fasting, blood from 23-month-old APP/PS1 and NTG mice that had been treated for 14 months with CP2 ($n = 15$–20 per group) or vehicle ($n = 15$–20 per group) was collected by orbital bleed and centrifuged for 10 min at 2500 rpm. Collected plasma was sent for 32-plex cytokine array analysis (Discovery Assay, Eve Technologies Corp. https://www.evetechnologies.com/discovery-assay/). The multiplexing analysis was performed using the Luminex 100 system (Luminex). More than 80% of the targets were within the detectable range (signals from all samples were higher than the lowest standard). For the data that were out of range (OOR<), their values were designated as 0 pg/ml. Blood samples were run in duplicates.

**Lipid peroxidation assay**. Levels of malondialdehyde (MDA), a product of lipid degradation that occurs as a result of oxidative stress, were measured using an MDA assay kit (#MAK085, Sigma Aldrich) in hippocampal brain tissue isolated from 23-month-old APP/PS1 and NTG mice treated for 14 months with CP2 or vehicle ($n = 5$ per group), according to the manufacturer's instructions.

**Levels of senescent cells in adipose tissue**. The senescent cell burden was assayed by senescent associated β-galactosidase (SA-β-Gal) staining of freshly isolated fat biopsies from 23-month-old APP/PS1 and NTG mice, treated for 14 months with CP2, or vehicle ($n = 5$–6 per group) as previously described[79]. In brief, about 100 mg of periovarian and inguinal fat pads were fixed in PBS containing 2.0% formaldehyde and 0.2% glutaraldehyde for 10 min. After fixation, tissue was washed and incubated with SA-β-gal staining solution for 16–18 h at 37 °C. The enzymatic reaction was stopped by washing tissue with ice cold PBS. Tissues were counterstained with DAPI, and ten to twelve random images were taken per sample with an EVOS microscope under 20x magnification in bright and fluorescent fields. SA-β-gal positive senescence cells and total cell number (DAPI+ nuclei) were quantified per field. SA-β-gal positive cell numbers were expressed as a percent of the total cell number per image. Representative images are included in the paper. Additional images are available upon request directed to Dr. E. Trushina.

**Metabolomics profiling for ceramide panel in blood**. Blood was collected from fasting (16 h) 23-month-old APP/PS1 and NTG mice treated for 14 months with CP2 or vehicle ($n = 5$ per group) by orbital bleeding. Concentrations of ceramides were established using targeted metabolomics at the Mayo Clinic Metabolomics Core using their established SOP as described previously[80].

**Metabolomics profiling in brain**. Fasting (12 h) APP/PS1 and NTG mice treated for 6 months with CP2 or vehicle ($n = 5$ per group) were sacrificed by cervical dislocation; brains were rapidly removed; hippocampal tissue was dissected and immediately flash-frozen in liquid $N_2$. Tissue was pulverized under liquid $N_2$ and extracted in a solution containing 0.6 M $HClO_4$ and 1 mM EDTA. Extracts were neutralized with 2 M $KHCO_3$ and used for metabolomic analysis as described in ref. [81]. An aliquot of 100 µl of extract was transferred into Eppendorf tube and spiked with 5 µl IS, myristic-d27 acid (1 mg/ml) at ambient temperature. Samples were gently vortexed and completely dried in a SpeedVac concentrator. The lyophilized brain samples were methoxiaminated and derivatized same way as for plasma samples. For GC-MS analysis, we used conditions previously optimized for an Agilent 6890 GC oven with Agilent 5973 MS[82]. Nucleotides were separated on a reversed-phase Discovery C18 columns (SIGMA, St. Louis, MO) with Hewlett-Packard series 1100 HPLC system (Agilent, Santa Clara, CA). The Agilent Fiehn

GC/MS Metabolomics RTL Library was employed for metabolite identification. GC-MS spectra were deconvoluted using AMDIS software. After that, SpectConnect software was used to create the metabolite peaks matrices.

**Electrophysiology**. APP/PS1 and NTG mice 15–18 months of age treated for ~10 months with CP2 or vehicle ($n = 5$ per group) were used for electrophysiology analysis. Mice were deeply anesthetized with isoflurane and decapitated. Brains were quickly removed and transferred into a cold slicing solution containing an artificial cerebrospinal fluid (ACSF), where NaCl was substituted with sucrose to avoid excitotoxicity. Transverse slices were made at 300–350 µm thickness using a vibratome (VT-100S, Leica). Slices were incubated in ACSF containing 128 mM NaCl, 2.5 mM KCl, 1.25 mM $NaH_2PO_4$, 26 mM $NaHCO_3$, 10 mM glucose, 2 mM $CaCl_2$, and 1 mM $MgSO_4$, aerated with 95% $O_2$/5% $CO_2$. Slices were maintained at 32 °C for 13 min, and then maintained at room temperature throughout the entire experiment. For electrophysiological recording, each slice (2–3 slices per mouse) was transferred to a recording chamber, and ACSF was continuously perfused at the flow rate of 2–3 ml/min. A single recording electrode and a single bipolar stimulation electrode were placed on top of the slice. A boron-doped glass capillary (PG10150, World Precision Instruments) was pulled with a horizontal puller (P-1000, Sutter Instrument) and filled with ACSF for extracellular recording. Under the microscope (FN-1, Nikon), the recording electrode was placed in the CA1 area of the hippocampus. The bipolar stimulation electrode (FHC) was placed at the Schaffer collaterals. The distance between two electrodes was over ~200 µm. To define a half response of stimulation, various intensities of electrical stimulation were applied (10–500 µA). However, the pulse width was fixed at 60 µs. Once the stimulation parameter was determined to generate a half maximum of evoked fEPSP, this stimulation intensity was used for the paired pulse and LTP experiments. The paired pulse stimulation protocol was applied with 50, 100, and 200 ms intervals. For the LTP experiment, test stimulation was applied every 30 s for 30 min to achieve a stable baseline. Once the stable baseline was achieved, a tetanic stimulation (100 Hz for 1 s) was applied three times at a 30-s intervals. Initial slopes of the fEPSP were used to compare synaptic strength. fEPSP slops were analyzed using pCLAMP v. 10.5 software.

**Mitochondrial DNA (mtDNA) copy number**. Genomic DNA was isolated from snap-frozen cortico-hippocampal brain sections from NTG and APP/PS1 mice treated with vehicle or CP2 ($n = 5$ per group) for 14 months using a DNeasy Blood and Tissue Kit (QIAGEN, cat. # 69504) according to the manufacturer's instructions. Quantification of mtDNA copy number was performed in triplicates using 100 ng of isolated DNA. qRT-PCR was performed using a iTaq™ DNA Polymerase (Bio-rad, cat. # 1708870) and primers for a genomic locus (*b-actin*) and a mitochondrial gene (*mND5*). Primers for *b-actin* were the following: forward sequence (5′-GAT CGA TGC CGG TGC TAA GA-3′); reverse sequence (5′-GGA AAA GAG CCT CAG GGC AT-3′). Primers for the amplification of *mND5*: forward sequence (5′-TGT AAA ACG ACG GCC AGT AGC CCT TTT TGT CAC ATG AT-3′); reverse sequence (5′-CAG GAA ACA GCT ATG ACC GGC TCC GAG GCA AAG TAT AG-3′). Thermal cycling protocol was as following: 10 min at 50 °C, 1 min at 95 °C, 10 s at 95 °C, 30 s at 60 °C. Cycles were repeated for 35 times. The relative mitochondrial DNA copy number was calculated as a ratio of genomic vs. mitochondrial DNA. Delta Ct (ΔCt) equals the sample Ct of the mitochondrial gene (*mND5*) subtracted from the sample Ct of the nuclear reference gene (*B-Actin*).

**Two-dimensional transmission EM (2D TEM)**. Hippocampal tissue from APP/PS1 and NTG mice treated with CP2 or vehicle for 12–14 months ($n = 5$ per group) was dissected, cut into 1 mm thick sections, and fixed in 4% paraformaldehyde +1% glutaraldehyde in 0.1 M phosphate buffer for 24 h. Fixed tissue was further cut into smaller (1 mm$^3$) pieces and placed into 8 ml glass sample vials (Wheaton, cat # 225534). Processing was facilitated by the use of a Biowave® laboratory microwave oven set to 150 W (Ted Pella Inc., Redding, CA) and included the following steps: (1) 0.1 M phosphate buffer (PB), pH 7.0, 40 s on, 2 min rest at RT, repeat three times; (2) 2% osmium tetroxide in $H_2O$, 40 s on, 40 s off, 40 s on, 15 min rest at RT; (3) $H_2O$ rinse, 40 s on, 2 min rest at RT, repeat three times; (4) 2% aqueous uranyl acetate, 40 s on, 40 s off, 40 s on, 15 min rest at RT; (5) $H_2O$ rinse, 40 s on, 2 min rest at RT, repeat three times; (6) sequential dehydration in ethanol series 60%, 70%, 80%, 95%, 100%, 100% acetone, 100% acetone, 40 s on, 2 min rest at RT each. Resin infiltration steps were performed as follows: (1) 1:2 resin:acetone, 40 s on, 40 s off, 40 s on, 15 min rest with the vacuum; (2) 1:1 resin:acetone, 40 s on, 40 s off, 40 s on, 15 min rest with the vacuum; (3) 3:1 resin: acetone, 40 s on, 40 s off, 40 s on, 30 min rest with the vacuum; (4) 100% resin 40 s on, 40 s off, 40 s on, overnight rest with the vacuum. The next day samples were moved to fresh resin in embedding molds and incubated at 60 °C for 24 h to polymerize. Ultrathin sections (silver interference color, ~0.1 µm) were place on 150 mesh copper grids and stained with lead citrate. Images were acquired using a JEOL 1400 + TEM operating at 80 kV equipped with a Gatan Orius camera. For conventional 2D TEM, approximately 50 images were taken randomly within each section for the analysis. All images used for the analysis were taken at ×5000 magnification. For 2D TEM analysis of mitochondrial morphology, organelles were scored according to their appearance as elongated (>2 µm long) or small (circular

0.5 μm in diameter). Twenty random areas from each CA1 region were imaged, and only neuropils longer than 3 μm were selected for the analyses. Representative images are included in the paper. Additional images are available upon request directed to Dr. E. Trushina.

**Three-dimensional (3D) EM using serial block face scanning electron microscopy (SBFSEM)**. Images for 3D EM reconstructions were obtained using an ApreoVolumeScope (Thermo Fisher Scientific) electron microscope that combines an integrated serial block face microtome (SBF) and high-resolution field emission scanning (SEM) imaging. Hippocampal CA1 region was dissected from vehicle and CP2-treated NTG and APP/PS1 mice, cut into 2 mm³ pieces, and immersion-fixed in neutral 2.5% glutaraldehyde + 2.5% paraformaldehyde in 0.1 M cacodylate buffer + 2 mM CaCl₂. Following 24 h fixation, samples were processed using the following protocol based on the serial block-face method developed by the National Center for Microscopy and Imaging Research (La Jolla, CA; https://ncmir.ucsd.edu/sbem-protocol): (1) samples were rinsed 4 × 3 min in 0.1 M cacodylate buffer + 2 mM CaCl₂, (2) incubated in 2% osmium tetroxide in 0.15 M cacodylate buffer for 1.5 h rotating at RT, (3) incubated in 2% osmium tetroxide + 2% potassium ferrocyanide in 0.1 M cacodylate for 1.5 h rotating at RT, (4) rinsed in H₂O 4 × 3 min, (5) incubated in 1% thiocarbohydrazide (TCH) 45 min at 50 °C, (6) rinsed in H₂O 4 × 3 min, (7) incubated in fresh 2% osmium tetroxide in H₂O 1.5 h rotating at RT, (8) rinsed in H₂O 4 × 3 min, (9) incubated in 1% aqueous uranyl acetate overnight at 4 °C, (10) further incubation in uranyl acetate 1 h at 50 °C, (11) rinsed in H₂O 4 × 3 min, (12) incubated in lead aspartate 1 h at 50 °C, (13) rinsed in H₂O 4 × 3 min, (14) dehydrated through ethanol series (60, 70, 80, 95, 100, 100%) 10 min each, (15) two rinses in 100% acetone 10 min each, (16) resin 1:2, 1:1, 3:1 in acetone 0.5 h, 1 h, 2 h, overnight in 100% resin. Samples were embedded into the Durcapan hard resin (EMS, Hatfield, PA), and allowed to polymerize at a minimum of 24 h prior to trimming and mounting. Tissue was trimmed of all surrounding resin and adhered to 8 mm aluminum pins (Ted Pella Inc., Redding, CA) using EpoTek silver epoxy (EMS, Hatfield, PA). A square tower (0.5 mm) was trimmed from the tissue using a Diatome ultratrim knife (EMS, Hatfield, PA) and the entire pin was coated with gold palladium. Following coating, the block was trimmed to a planer surface using a diamond knife, and the pin was mounted in the SEM microtome. Serial block-face images were acquired using a Thermo Fisher Volumescope (Thermo Fisher, Inc., Waltham, MA) at 50 nm section depths with detector acceleration voltage of 1.5 kV. Voxel size ranged between 8 and 14 nm according to magnification. For each region of interest (ROI), 400 sections were obtained, registered, and filtered using non-local means. All images used for the analysis were adjusted to ×5000 magnification. Segmentation and three-dimensional analysis was performed using *Reconstruct* (SynapseWeb)[83] and *Amira* 6.4 software (Thermo Fisher, Inc., Waltham, MA).

**Image segmentation and quantitative morphometric analysis of dendritic spines using 3D EM**. Aligned and normalized stacks of serial sections were further processed with unsharp masking, Gaussian blur and non-local means filters were applied in order to clearly distinguish cellular membranes. Dendrites, dendritic spines and synapses (the PSD and the opposed presynaptic membranes) were segmented by manually tracing contours in consecutive micrographs. Each segmented dendritic spine or synaptic junction was identified independently. Once the dendrite reconstruction was completed in *Reconstruct*, traces were modified by an absolute intensity and maximal thresholding approach and exported in JPEG format into *Amira* software. The magic wand tool in *Amira*'s segmentation mode was used to pull out the reconstructions from the *Reconstruct* exports. Dendritic spines were cut from their parent dendritic shaft through the base of their neck in a 3D optimal orientation and analyzed using the *label analysis* function in *Amira*'s *project view*. The label analysis was customized to measure 3D length, surface area, and volume of the spines. By using the *measure tool* in *Amira*'s project view, length, width of the head and length of the neck of each dendritic spine were established. The following classifications for dendritic spine type were adapted from[40]: branched (2 or more heads); filopodium (length > 2 μm, no bulbous head); mushroom (length 1 < x > 2 μm; head > 0.6 μm); long thin (length 1 < x < 2 μm; head < 0.6 μm); thin (length < 1 μm; head < 0.6 μm); and stubby (length < 1 μm; length: width ratio < 1). In addition to the frequency of individual spine types, their activity (based on the presence of synaptic vesicles), volume and size (length and width) of spines were also estimated. To evaluate the impact of nanoscale alterations in spine morphology on the diffusional coupling between spines and dendrites that promote long-term potentiation, we calculated the compartmentalization factor, which is defined as $CF = V \times L/A$ where $V$ is the spine head volume, $L$ is the spine neck length, and $A$ is the cross-sectional area of the spine neck[42]. Representative images are included in the paper. Additional images are available upon request directed to Dr. E. Trushina.

**Next-generation RNA sequencing**. Brain tissue, encompassing the hippocampal and cortical regions, from APP/PS1 and NTG mice treated with vehicle of CP2 for 14 months (n = 5 *per* group) were lysed in QIAzol (Qiagen cat. # 79306) followed by RNA isolation using miRNeasy (Qiagen cat. # 217004) according to the manufacturer's instructions. The quantity and quality of RNA were measured using a NanoDrop spectrophotometer and Agilent 2100 Bioanlyzer, respectively. All RINs (RNA integrity numbers) had a value greater than eight.

*Library preparation and sequencing*. Total RNA (200 ng) was used to generate libraries using TruSeq RNA Library Prep Kit v2 (Illumina). All samples were sequenced at the Mayo Clinic Medical Genome Facility (MGF) Sequencing Core by Illumina HiSeq 4000 with paired end 101-bp read length. Approximately 50 million single fragment reads were acquired per sample.

*Bioinformatics methods*. MAP-RSeq v 2.1.1, a comprehensive computational pipeline developed by the Mayo Clinic's Division of Biomedical Statistics and Informatics, was used to analyze RNA-Sequencing data. MAP-RSeq uses a variety of publicly available bioinformatics tools tailored by methods developed in-house. The main outputs of the MAP-RSeq workflow are the gene counts, expressed single nucleotide variants (eSNVs), gene fusion candidates, and quality control plots. The aligning and mapping of reads was performed using TopHat2 against the mm10 reference genome. The gene and exon counts were generated by FeatureCounts using the gene definitions files from Ensembl. '-O' option within FeatureCounts was used to account for expression derived from regions shared by multiple genomic features. FeatureCounts was also executed for quantifying expression on a per exon basis by utilizing the '-f' option. RSeqQC[84] was used to create a variety of quality control plots to ensure the results from each sample were reliable and could be collectively used for differential expression analysis. Upon the completion of all the deliverables from MAP-RSeq, a html document was created tying everything together in one interactive document. The R bioinformatics package DeSeq2 was used for differential gene expression analysis. The criteria for the selection of significant differentially expressed genes was *P* value < 0.05. Gene function enrichment was determined using the Database for Annotation, Visualization and Integrated Discovery (DAVID v. 6.8)[85]. Mouse transcriptional factors were downloaded from the Riken genome database and mapped to our RNA-seq differential gene list. The clustering heatmaps were generated based on unsupervised hierarchical clustering using Pearson correlation distance. Most graphs were generated using customized R programs. RNA seq data availability: (GEO accession ID is GSE149248). Code (R script) used to generate final RNA seq analysis is available in Supplementary Data 22, 23.

**Mapping mouse genes to human AD genes from the AMP-AD study**. The differentially expressed genes from our RNA-seq data using mouse model were identified as described in the differential analysis section. The upregulated and downregulated genes in comparison of control and AD females were from the AMP-AD data set. The human data and analysis process were described in Wan et al.[86]. The upregulated genes in the RNA-seq data of the mouse NTG vs. AD comparison were overlapped with the human upregulated gene list. The downregulated genes in mouse data were overlapped with the human downregulated list. Gene function enrichment analysis was performed on the overlapped list of DEGs using NCBI DAVID function analysis tools with a *P* value cutoff of less than 0.05. Similar analyses were performed on the 567 genes overlapped in the two differential gene lists of NTG vs. AD and AD vs. AD + CP2 comparisons. All genes are presented in the Supplementary Data 2–19.

**Comprehensive laboratory animal monitoring system (CLAMS)**. The CLAMS (Columbus Instruments, Columbus, OH) allows automated, non-invasive, and simultaneous monitoring of horizontal and vertical activity, feeding and drinking, oxygen consumption, and CO₂ production of an individual mouse. APP/PS1 and NTG mice treated with vehicle or CP2 (16–18-month-old, n = 15–20 per group) were individually placed in CLAMS cages. Indirect calorimetry was monitored over 2 days, when mice were allowed food for 24 h ad lib (Fed state), and for the following 24 h food was removed (Fasting state). Mice were maintained at 20–22 °C under a 12:12 h light–dark cycle. All mice were acclimatized to CLAMS cages for 3–6 h before recording. Sample air was passed through an oxygen sensor for the determination of oxygen content. Oxygen consumption was determined by measuring oxygen concentration in the air entering the chamber compared with air leaving the chamber. The sensor was calibrated against a standard gas mix containing defined quantities of oxygen, carbon dioxide, and nitrogen. Food and water consumption were measured directly. The hourly file displayed measurements for the following parameters: VO₂ (volume of oxygen consumed, ml/kg/h), VCO₂ (volume of carbon dioxide produced, ml/kg/h), RER (respiratory exchange ratio), heat (kcal/h), total energy expenditure (TEE, kcal/h/kg of lean mass), activity energy expenditure (AEE, kcal/h/kg of lean mass), resting total consumed food (REE kcal/h/kg of lean mass), food intake (g/kg of body weight/12 h), metabolic rate (kcal/h/kg), total activity (all horizontal beam breaks in counts), ambulatory activity (minimum 3 different, consecutive horizontal beam breaks in counts), and rearing activity (all vertical beam breaks in counts). The RER and atty acid (FA) oxidation were calculated using the following equations: $RER = VCO_2/VO_2$; FA oxidation (kcal/h) = TEE × (1 − RER/0.3). Daily FA oxidation was calculated from the average of 12 h of hourly FA oxidation. Daily carbohydrate plus protein oxidation was calculated from average of 12 h of hourly TEE minus daily FA oxidation. Metabolic flexibility was evaluated from the difference in RER between

daily fed state and fasted state recorded at night phase, according to the following equations: $\Delta = 100\% \times$ (RER fed–RER fasted)/RER fed.

**Dual energy X-ray absorptiometry (DEXA).** A LUNAR PIXImus mouse densitometer (GE Lunar, Madison, WI), a dual-energy supply X-ray machine, was used for measuring skeletal and soft tissue mass for the assessment of skeletal and body composition in CP2 or vehicle-treated mice. Live mice were scanned under 1.5–2% isoflurane anesthesia. Mice were individually placed on plastic trays, which were then placed onto the exposure platform of the PIXImus machine to measure body composition. The following parameters were generated: lean mass (g), fat mass (g), and the percentage of fat mass. These parameters were used to normalize data generated in CLAMS, including $O_2$, $VCO_2$, metabolic rate and energy expenditure.

**In vivo $^{31}P$ NMR spectroscopy.** NMR spectra were obtained in 16–18-month-old APP/PS1 and NTG mice treated with CP2 or vehicle for 8 months ($n = 5$ per group) using an AVANCE III 300 MHz (7 T) wide bore NMR spectrometer equipped with micro-imaging accessories (Bruker BioSpin, Billerica, MA) with a 25-mm inner diameter dual nucleus ($^{31}P/^1H$) birdcage coil. For anatomical positioning, a pilot image set of coronal, sagittal, and axial imaging planes were used. For $^{31}P$ NMR spectroscopy studies, a single pulse acquisition with pulse width: P1 200 us (~30°), spectral width: SW 160 ppm, FID size: TD 16 k, FID duration: AQT 0.41 s, waiting time: D1 1 s and number of scans: NS 512 was used. The acquisition time was 12 min. Spectra were processed using TopSpin v1.5 software (BrukerBiospin MRI, Billerica, MA). Integral areas of spectral peaks, corresponding to inorganic phosphate (Pi), phosphocreatine (PCr), and γ, α, and β phosphates of adenosine triphosphate (αATP, βATP, and γATP), were analyzed using jMRUI software. Since in some mice, the Pi peaks were not detectable, the Pi values were uniformly omitted from the analyses. The presence of phosphomonoester (PME) or phosphodiester (PDE) peaks was also recorded. However, the signal-to-noise ratio of these peaks was not always adequate for accurate quantification. Levels of PCr and Pi were normalized to the total ATP levels present in that spectrum or to the amount of βATP. Results were consistent between both of these normalization methods; data are presented as the ratio of each parameter to total ATP.

**Western blot analysis.** Levels of proteins were determined from the cortico-hippocampal region of the brain of vehicle and CP2-treated NTG and APP/PS1 mice ($n = 6–8$ mice per group) by Western blot analysis. Tissue was homogenized and lysed using 1× RIPA Buffer plus inhibitors. Total protein lysates (25 μg) were separated in equal volume on 4–20% Mini-PROTEAN TGX™ Precast Protein Gels (Bio-Rad, cat. # 4561096) and transferred to an Immun-Blot polyvinylidene difluoride membrane (PVDF cat. # 1620177). The following primary antibodies were used: phospho-AMPK (Thr 172) (1:1000, Cell Signaling Technology, cat. # 2535), AMPK (1:1000, Cell Signaling Technology, cat. # 2532), phospho-Acetyl-CoA Carboxylase (Ser79) (1:1000, Cell Signaling Technology, cat. # 11818), Synaptophysin (1:200, Santa Cruz Biotechnology, Santa Cruz, CA, cat. # 17750), BDNF (1:200, Santa Cruz Biotechnology, Santa Cruz, CA, cat. # 546), PSD95 (1:1000, Cell Signaling Technology, cat. # 2507), Sirt3 (1:1000, Cell Signaling Technology, cat. # 5490), IGF1 (1:2000, Abcam, cat. # ab223567), Superoxide Dismutase 1 (1:1000, Abcam, cat. # ab16831), phospho-IGF-I Receptor β (Tyr1131)/Insulin Receptor β (Tyr1146) (1:1000, Cell Signaling Technology, cat. # 3021), IGF-I Receptor β (1:1000, Cell Signaling Technology, cat. # 3027), phospho-Pyruvate Dehydrogenase α1 (Ser293) (1:1000, Cell Signaling Technology, cat. # 31866), Pyruvate Dehydrogenase (1:1000, Cell Signaling Technology, cat. # 3205), Glut4 (1:1000, Cell Signaling Technology, cat. # 2213), Glut3 (1:1000, Santa Cruz Biotechnology, cat. # sc-74399), phospho-ULK1 (Ser555) (1:1000, Cell Signaling Technology, cat. # 5869), phospho-ULK1 (Ser317) (1:1000, Cell Signaling Technology, cat. # 12753), phospho-Beclin-1 (Ser15) (1:1000, Cell Signaling Technology, cat. # 84966), PGC1α (1:1000, Calbiochem, cat. # KP9803), TFAM (1:1000, Sigma-Aldrich, cat. # AV36993), phospho-NF-κB p65 (Ser536) (1:1000, Cell Signaling Technology, cat. # 3033), HO-1 (1:1000, Cell Signaling Technology, cat. # 70081), IκBα (1:1000, Cell Signaling Technology, cat. # 4812), Nrf2 (1:1000, Abcam, cat. # ab62352), LC3B (1:1000, Novus Biologicals, cat. # NB100-2220), TFEB (1:500, Thermo Fisher, cat. # PA5-75572), LAMP1 (1:1000, Cell Signaling Technology cat. # 9091), Catalase (1:1000, Cell Signaling Technology cat. # 14097), Tubulin (1:5000, Biovision, cat. # 3708), β-Actin (1:5000, Sigma-Aldrich, cat. # A5316). The following secondary antibodies were used: donkey anti-rabbit IgG conjugated with Horseradish Peroxidase (1:10,000 dilution, GE Healthcare UK Limited, UK) and sheep anti-mouse IgG conjugated with Horseradish Peroxidase (1:10,000 dilution, GE Healthcare UK Limited, UK). Ban quantification was done using Image Lab™ v. 6.0.

**Statistics and reproducibility.** The statistical analyses were performed using the GraphPad Prism (Version 8, GraphPad Software, Inc., La Jolla, Ca). Statistical comparisons among four groups concerning behavioral and metabolic tests, immunoreactivity, metabolomics, plasma cytokine panel, body composition, electron microscopy imaging, FDG-PET, $^{31}P$ NMR, electrophysiology, were analyzed by two-way ANOVA, the two-tailed unpaired and paired Student $t$ test, where appropriate. The Fisher's LSD post hoc analysis was used if significant interaction among groups was found. A linear regression analysis was applied to determine differences among the groups in body weights and age-related loss of TH+ axons and neurons. Significant differences between vehicle and CP2-treated groups within the same genotype and differences among NTG, APP/PS1, and APP/PS1 + CP2 mice were considered in the final analysis. Data are presented as mean ± S.E. M. for each group of mice. All $P$ values generated in this study are presented in Supplementary Data 20, and individual values used to generate plots in the manuscript are presented in Supplementary Data 21. In reporting the outcomes of the IPIST experiments, 1 NGT+CP2 and 2 APP/PS1+CP2 mice were excluded from the graph based on the application of the Grubb's test to identify the outliers. Sample sizes were determined by setting a minimum $n$ number for in vitro biological replicates at 3, to allow for statistical testing, however in most cases $n$ numbers were higher. All replicates displayed in this paper are biological replicates, technical replicates (usually 3) were performed and used to generate the means for each biological replicate. At the initiation of each experiment, all available data from previous studies were assessed and used to adequately design the experiment to have at least 80% power to detect a biologically meaningful effect size while controlling the type I error rate at 5%. Animal allocation to treatment groups was randomized. We were blinded to both the genetic and treatment information, being unblinded after analysis was complete.

## Data availability

RNA-seq data generated in this study using mice that support the findings are deposited to public repository and are freely available at: (GEO accession ID is GSE149248). Source data underlying plots shown in figures are presented in Supplementary Data 21. Full blots are available in Supplementary Information.

## Code availability

Code (R script) used to generate final RNA seq data analysis is available in Supplementary Data 22 and 23.

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

## Acknowledgements

We thank Mayo Clinic Cores for help with FDG-PET; RNA-seq, metabolomics, EM and CLAMS data acquisition; Dr. A. Leontovich, Ms. R. Schlichte, E. Murray, L. G. Andres-Beck, and Mr. I. Trushin for help. This research was supported by grants from the National Institutes of Health NIA RF1AG55549 (to E.T.), NINDS R01NS107265 (to E. T.), RO1AG062135 (to E.T. and M.K.L.), ADDF 291204 (to E.T.), MN Partnership for Biotechnology and Medical Genomics # 15.08 (to E.T. and M.K.L.), NIH RO1NS88260 (to S.Y.C.), NIH RO1 NS112381 (to A.G.), National Cancer Institute Grant P30 CA015083 (to J.M.R.), NIH grants R37AG013925 and P01AG062413 (to J.L.K. and T.T.), the Alzheimer's Association Part the Cloud Program, Robert and Arlene Kogod, the Connor Group, Robert J. and Theresa W. Ryan, and the Noaber Foundation. Its contents are solely the responsibility of the authors and do not necessarily represent the official view of the NIH. The funders had no role in study design, data collection and analysis, decision to publish, or preparation of the manuscript.

## Author contributions

E.T. conceived the study, assembled the multidisciplinary team of collaborators, and with M.L. received funding for the project. A.S., S.T., M.L., and E.T. performed the research, analyzed and interpreted data, and wrote the paper; X.L. and M.D. provided bioinformatics assistance for RNA-seq data analysis; C.F. provided data from the human AMP-AD RNA data set; X.L. and A.S. analyzed and interpreted transcriptome data and contributed to corresponding figures and tables; A.S., P.F., B.G., J.N., X.Z., U.T., and J.W. performed in vitro and in vivo studies in mice, tissues and biofluids under the supervision of E.T.; T.C. and A.J. prepared brain tissue for EM under the supervision of J.L.S.; L.K and R.E.G. analyzed the EM data under the supervision of A.S. and E.T.; A.S. and C. Y.I.K. conducted unbiased stereology and immunohistochemistry in brain for neurodegeneration and inflammation assessment under the supervision of M.K.L.; S.Y.C. and A.S. conducted electrophysiology experiments; Anna S. and T.O. conducted experiments with complex I inhibition under the supervision of A.G.; S.M. assisted with 31 P NMR; T. T., T.P., and J.L.K. conducted the evaluation of senescent burden; R.A.K. and R.A.S. conducted PK studies under the supervision of J.M.R.; Y.Y. established levels of Aβ peptides under the supervision of T.K.; S.Z. and E.N. conducted metabolomic analysis under the supervision of P.D. All authors edited the manuscript and approved its publication.

## Competing interests

The authors declare no competing interests.

## Additional information

[1]Department of Neurology, Mayo Clinic, 200 First St. SW, Rochester, MN 55905, USA. [2]Institute for Translational Neuroscience, University of Minnesota Twin Cities, 2101 6th Street SE, Minneapolis, MN 55455, USA. [3]Department of Neurologic Surgery, Mayo Clinic, 200 First St. SW, Rochester, MN 55905, USA. [4]Department of Physiology and Biomedical Engineering, Mayo Clinic, 200 First St. SW, Rochester, MN 55905, USA. [5]Division of Biomedical Statistics and Informatics, Department of Health Sciences Research, Mayo Clinic, 200 First St. SW, Rochester, MN 55905, USA. [6]Microscopy and Cell Analysis Core, Mayo Clinic, 200 First St. SW, Rochester, MN 55905, USA. [7]Department of Biochemistry and Molecular Biology, Mayo Clinic, 200 First St. SW, Rochester, MN 55905, USA. [8]Institute for Systems Biology, Seattle, WA 98109-5263, USA. [9]Division of Neonatology, Department of Pediatrics, Columbia University, 116th St & Broadway, New York, NY 10027, USA. [10]Robert and Arlene Kogod Center on Aging, Mayo Clinic, 200 First St. SW, Rochester, MN 55905, USA. [11]Department of Molecular Pharmacology and Experimental Therapeutics, Mayo Clinic, 200 First St. SW, Rochester, MN 55905, USA. [12]Department of Neuroscience, Mayo Clinic, 4500 San Pablo Road, Jacksonville, FL 32224, USA. [13]Department of Cardiovascular Medicine, Mayo Clinic, 200 First St. SW, Rochester, MN 55905, USA. [14]Faculty of Pharmacy, Department of Analytical Chemistry, Hacettepe University, Sihhiye, Ankara 06100, Turkey. [15]These authors contributed equally: Andrea Stojakovic, Sergey Trushin, Anthony Sheu. [16]These authors jointly supervised this work: Michael K. Lee, Eugenia Trushina.
✉email: trushina.eugenia@mayo.edu

