## [Peer Review File · Communications Biology]

Reviewers' comments:

Reviewer #1 (Remarks to the Author):

Stojakovic et al. present a manuscript entitled "Partial inhibition of mitochondrial complex I attenuates neurodegeneration and restores energy homeostasis and synaptic function in a symptomatic Alzheimer's mouse model" in which they describe the treatment of a transgenic mouse model of Alzheimer's disease (AD) with the tricyclic pyrone, CP2. The manuscript is well written and very comprehensive. The authors demonstrate how CP2 improves cognition and motor function in APP/PS1 mice (a common translational model of AD) and back up their findings with a myriad of tests. In addition, the authors continue to demonstrate how this potential treatment increases neuroprotection through a variety of metabolic pathways to include improved homeostasis, reduced A-beta accumulation and toxicity, reduced oxidative stress etc. all through partial inhibition of mitochondrial respiratory chain complex I. The data presented is quite compelling and does have potential translational applications for the treatment of AD based on their recapitulation of results using previously acquired data (AMP-PD transcriptomic data). Overall, I found the manuscript to be very interesting and novel. However, I do have some points which will need clarification and/or additional follow up studies.

Major:

All of this work is completed in an A-beta mouse model of AD and has not made any considerations for the effect of tauopathy on the progression of the disease. The authors rightly state that a lot of previous work has been undertaken to study potential therapeutics aimed at A-beta accumulation and problems have arisen for the listed reasons. While some mention of how these results may relate to pTau in the brain of AD sufferers (page 28), I believe further information is required.

The title is also a little misleading. Only female mice were used in this study and the data compared to female humans. Numerous studies have been published (in the brain in particular) noting the differences in the "omes" between male and females. This is particularly important considering the number of omics technological platforms used herein. As such this is not a study of Alzheimer's disease but of how MCI inhibition in females may confer some neuroprotection and increase in cognition. I would suggest analyzing male mice and comparing the transcriptome with male human data, but this might be beyond the scope of this investigation now.

I would also suggest the authors include a limitations section in their Discussion section noting these.

Minor:

How many mice were used is not clear for each study. A range is provided, and I was wondering why.

Provide the IACUC # for the study performed at the Mayo Clinic.

Reviewer #2 (Remarks to the Author):

The manuscript by Stojakovic et al. efficiently demonstrated that mitochondrial respiratory chain complex I is a potential therapeutic target for normalization of clinical/biochemical features associated to Alzheimer's disease. The paper is well written, experiments well-performed with

appropriate controls and methods are adequately described. Statistical analyses of this data are pretty standard, although I included couple points below. In my view, there are some minor issues that need to be addressed for a better reading experience of this manuscript before it can be accepted and published. The issues are described below -

1. In figure 1B, CP2 treatment seems to increase weight in general. Did the authors notice a difference in the food intake in this animals? Could this observation may affect/explain some of the metabolic changes that the authors noticed in the work?

2. In case of nomenclature, I suggest adhering to the commonly used names of neurotransmitters - gamma-aminobutyrate in place of 4-aminobutyrate - that may be more familiar to the general reader.

3. I suggest authors be more careful in interpretation of their metabolomic data. While their general picture fits well with elevated anabolism hypothesis, elevation in certain amino acids may not fit with that. For example, beta-alanine is a catabolic product (please refer to human metabolome database/KEGG). Also, amino acids such as valine, serine and glycine are either essential or conditionally essential amino acids. Their elevation may not necessarily reflect altered anabolism/synthesis. The authors may want to rewrite this interpretation.

4. Statistics -

A. The authors should identify the explained variances by PCs in their principal component analysis plot. B. The authors may want to consider providing the exact p-values in their manuscript. I understand that can be problematic in the figure, perhaps they can include a separate supplementary file. C. The regression plots should include the R-squared and p-values at the minimum and probably the parameter estimates too.

Response to Reviews

We are very grateful for constructive comments provided by the Reviewers of our manuscript. Addressing these questions undoubtedly brought better clarity to our work, which by all Reviewers was found “*well written and very comprehensive, compelling, with potential translational applications, interesting and novel*”.

Our point-by-point response is provided below.

Reviewer 1

1. The Reviewer requested additional clarification of how treatment with MCI inhibitors may affect the pTau pathology. The Reviewer also asked about potential sex-specific differences associated with the treatment efficacy.

We are very happy to report that the manuscript entitled “***Partial Inhibition of Mitochondrial Complex I Reduces Tau Pathology and Improves Energy Homeostasis and Synaptic Function in 3xTg-AD Male and Female Mice***” was recently reviewed in the *Journal of Alzheimer’s Disease* and is under resubmission. This manuscript specifically addresses efficacy of CP2 treatment in 3xTg-AD male and female symptomatic mice. This model expresses mutant human APP and PS1 proteins (responsible for a production of human A β peptides) together with the mutant human Tau protein resulting in the accumulation of human pTau and altered synaptic transmission and LTP. This model is considered the best available to recapitulate synergistic A β and Tau pathologies. In this manuscript, we provided compelling evidence that CP2 treatment restores glucose uptake and utilization, dendritic spine function, LTP and cognitive function in male and female 3xTg-AD mice. These parameters were specifically interrogated since translocation of pTau to dendritic spines and its interference with synaptic function and LTP are among the pTau-specific mechanisms of AD. Our findings indicate that partial inhibition of MCI in 3xTg-AD mice induced beneficial effect similar to the results presented in the current manuscript submitted to Communications Biology. Our results generated in 3xTg-AD mice demonstrated that chronic CP2 treatment was efficacious in both male and female mice. We added the following text to the Discussion:

Indeed, our independent study in the mouse model of AD that along with the mutant human APP and PS1 proteins also expresses mutant human Tau protein, the 3xTg-AD mice⁶⁹, demonstrated that chronic CP2 treatment reduced pTau levels and improved LTP and energy homeostasis in symptomatic male and female mice. While the current study interrogated CP2 efficacy only in female APP/PS1 mice, treatment in 3xTg-AD mice was beneficial in males and females. These data further support the concept that targeting mitochondria with small molecule specific MCI inhibitors represents a promising strategy that could be efficacious in patients with synergistic A β - and pTau-related pathology.

We also referred to the limitation of this study by including in the discussion (p.18) the following statement:

However, this study was limited only to female mice.

2. Based on the Reviewer's suggestions, we change the title to reflect that the study was done in female APP/PS1 mice only. The new title is "*Partial inhibition of mitochondrial complex I attenuates neurodegeneration and restores energy homeostasis and synaptic function in symptomatic APP/PS1 female mice*"
3. We clarified the number of mice in each study in Figure legends.
4. We added the Mayo Clinic IACUC number to the Method section.

Reviewer 2 requested the clarification of four points:

1. *In figure 1B, CP2 treatment seems to increase weight in general. Did the authors notice a difference in the food intake in this animals? Could this observation may affect/explain some of the metabolic changes that the authors noticed in the work?*

To clarify this issue, we included data on food intake and metabolic rate collected within the Metabolic study (CLAMS) in Supplementary Figure 3. We included this paragraph in the Result section:

Data generated using indirect calorimetry (CLAMS) demonstrated that CP2- and vehicle-treated NTG mice had similar food intake (Supplementary Figure 3b). However, CP2 treatment in NTG mice decreased the metabolic rate at rest (Supplementary Figure 3c). Since general daily activity of vehicle- and CP2-treated NTG mice was not different (Figure 2f), the significant weight gain observed in CP2-treated NTG mice could be associated with reduced metabolic rate and lower energy expenditure. In comparison to NTG mice, vehicle-treated APP/PS1 mice displayed lower body weight, higher food intake, increased metabolic rate and motor activity (Supplementary Figure 3a-c, Figure 2f). This is consistent with previous findings showing that AD mice have elevated caloric intake, hypermetabolism and lower body weight²⁵. The weight gain in CP2-treated APP/PS1 could be explained by increased food intake and attenuated hyperactivity, while metabolic rate remained similar between CP2- and vehicle-treated groups (Figure 2b,f, Supplementary Figure 3a-c). Since CP2 treatment did not increase food intake in NTG mice, it is feasible that increased food intake in CP2-treated APP/PS1 mice could be related to more frequent feeding due to ameliorated hyperactivity.

2. *In case of nomenclature, I suggest adhering to the commonly used names of neurotransmitters - **gamma-aminobutyrate** in place of 4-aminobutyrate - that may be more familiar to the general reader.*

We corrected this in the text and in Supplementary Table 5.

3. *I suggest authors be more careful in interpretation of their metabolomic data. While their general picture fits well with elevated anabolism hypothesis, elevation in certain amino acids may not fit with that. For example, beta-alanine is a catabolic product (please refer to human metabolome database/KEGG). Also, amino acids such as valine, serine and glycine are either essential or conditionally essential amino acids. Their elevation may not necessarily reflect altered anabolism/synthesis. The authors may*

want to rewrite this interpretation.

We limited the discussion on metabolic changes to the ones that were supported by other results in this manuscript. The new discussion is as following:

In agreement with target engagement, CP2 treatment increased levels of AMP in APP/PS1 mice but did not reduce brain levels of ATP. Treatment resulted in increased levels of citrate and N-acetyl aspartate (NAA), markers of improved mitochondrial and neuronal function. Importantly, levels of 2-hydroxyglutarate, a marker of detrimental mitochondrial stress²⁹, were not elevated, suggesting inhibition of MCI with CP2 does not induce adverse mitochondrial stress that could negatively regulate neuronal survival and function. Increased levels of gamma-aminobutyrate, a metabolite involved in the gamma-aminobutyric acid (GABA) neurotransmitter system, indicate an improvement in neurotransmission and potential reversal of AD-related neurodegeneration. Increased levels of ascorbic/dehydroascorbic acids imply an improvement in vitamin C status and redox balance in the brain, consistent with CP2-induced activation of neuroprotective mechanisms (Figure 1d-l).

4. Statistics -

A. The authors should identify the explained variances by PCs in their principal component analysis plot.

We identified and included the following sentence in the Supplementary Figure 13a: *“Principal Component Analysis (PCA) shows separated clusters of samples among 3 groups (NTG, green; APP/PS1, blue; APP/PS1+CP2, orange) where CP2 treatment produces specific changes allowing group separation. Variances explained by the first two principal components are shown here (15.37% and 13.35%, respectively)”* and included the values on the PCA plot.

B. The authors may want to consider providing the exact p-values in their manuscript. I understand that can be problematic in the figure, perhaps they can include a separate supplementary file.

The exact p-values for data generated in all Figures associated with this paper were added to Supplementary Data 19.

C. The regression plots should include the R-squared and p-values at the minimum and probably the parameter estimates too.

We included R- squared and p- values in the figure legends of the respective figures where data were analyzed by linear regression analysis. Those figures are: Figure 2b, Supplementary Figure 12c,d.

REVIEWERS' COMMENTS:

Reviewer #1 (Remarks to the Author):

The authors have addressed all of my concerns satisfactorily.

Reviewer #2 (Remarks to the Author):

In the revised version of the manuscript, I find all of my concerns sufficiently addressed. Therefore, I believe this manuscript may be published in the current form.